# Structural properties of a haemophore facilitate targeted elimination of the pathogen *Porphyromonas gingivalis*

Jin-Long Gao[1,2], Ann H. Kwan[3], Anthony Yammine[3], Xiaoyan Zhou[1], Jill Trewhella [3], Barbara M. Hugrass[4], Daniel A.T. Collins[4], James Horne[5], Ping Ye[2], Derek Harty[2], Ky-Anh Nguyen[1,2], David A. Gell [4] & Neil Hunter[1,2]

*Porphyromonas gingivalis* is a keystone bacterial pathogen of chronic periodontitis. *P. gingivalis* is unable to synthesise the porphyrin macrocycle and relies on exogenous porphyrin, including haem or haem biosynthesis intermediates from host sources. We show that under the iron-limited conditions prevailing in tissue environments, *P. gingivalis* expresses a haemophore-like protein, HusA, to mediate the uptake of essential porphyrin and support pathogen survival within epithelial cells. The structure of HusA, together with titration studies, mutagenesis and in silico docking, show that haem binds in a hydrophobic groove on the α-helical structure without the typical iron coordination seen in other haemophores. This mode of interaction allows HusA to bind to a variety of abiotic and metal-free porphyrins with higher affinities than to haem. We exploit this unusual porphyrin-binding activity of HusA to target a prototypic deuteroporphyrin-metronidazole conjugate with restricted antimicrobial specificity in a Trojan horse strategy that effectively kills intracellular *P. gingivalis*.

[1] School of Dentistry, Faculty of Medicine and Health, The University of Sydney, Camperdown, NSW 2145, Australia. [2] Institute of Dental Research, Westmead Centre for Oral Health, Westmead, NSW 2145, Australia. [3] School of Life and Environmental Sciences and Sydney Nano, The University of Sydney, Camperdown, NSW 2006, Australia. [4] School of Medicine, The University of Tasmania, Hobart, TAS 7000, Australia. [5] Central Science Laboratory, The University of Tasmania, Sandy Bay, TAS 7005, Australia. These authors contributed equally: Jin-Long Gao, Ann H. Kwan Correspondence and requests for materials should be addressed to J.-L.G. (email: jinlong.gao@sydney.edu.au) or to D.A.G (email: david.gell@utas.edu.au)

The Gram-negative anaerobe *Porphyromonas gingivalis* is a "keystone" pathogen in chronic periodontitis; whereby, the capacity of the organism to dysregulate local host defence mechanisms drives a shift in the dental plaque microbiota towards dysbiosis[1]. Subsequent host response against the dysbiotic microflora results in progressive destruction of supporting structures of the teeth and eventual tooth loss[2]. Critically, one strategy *P. gingivalis* employs to avoid immune surveillance is to invade oral epithelial cells[3]. High levels of intracellular *P. gingivalis* can be recovered from the oral cavity of subjects with periodontitis[4], and this provides a reservoir for recurrent infection after therapy[5]. Effective targeting of intracellular *P. gingivalis* is therefore an important aspect of enhanced antimicrobial therapy. Current treatment strategies, including conventional mechanical debridement in conjunction with systemic or local antibiotics are only partially effective in eliminating *P. gingivalis* and preventing recurrent infection[6]. In particular, the prolonged application of broad spectrum antibiotics required to kill intracellular latent bacteria could contribute to the development of drug resistance. Emergence of clinical isolates of *P. gingivalis* with antibiotic resistance genes[7,8] indicates the need for development of new antibiotics to achieve targeted control of *P. gingivalis*.

As a porphyrin auxotroph, *P. gingivalis* has highly adapted systems for acquiring haem from host haemoproteins[9]. Haem uptake system protein A (HusA) is a haemophore-like protein that is essential for *P. gingivalis* growth under the haem-limited conditions that prevail in tissue environments[10]. Antibodies reactive with HusA are found in patients with chronic periodontal disease, supporting the contention that HusA is expressed during infection[10]. A BLASTP search shows that HusA homologues are limited to the phylum *Bacteroidetes*, which includes many free-living species. Highly homologous sequences, however, are present only in the genus *Porphyromonas* (Supplementary Figure 1). These properties raise the possibility that HusA might be exploited to deliver an antibiotic cargo with high species-specificity by coupling to a porphyrin moiety.

Here we show that expression of HusA is essential for the intracellular survival of *P. gingivalis*. We present the solution structure of HusA and, from titration and mutagenesis studies, map the haem/porphyrin-binding site to a hydrophobic groove. Of note, HusA binds to a variety of porphyrins, including abiotic deuteroporphyrin IX (DPIX), with significantly higher affinities than it binds to haem. We use this property to deliver a deuteroporphyrin-metronidazole antibiotic as a "Trojan horse", resulting in effective elimination of intracellular *P. gingivalis*.

## Results

**P. gingivalis requires HusA under iron-limited growth**. To explore the role of HusA, a deletion mutant (ΔhusA) and a complementation mutant (ΔhusA+) were constructed[10] and, together with wild-type *P. gingivalis* strain W83, these were assessed for growth on various haem/porphyrin sources. As the iron normally present in host tissues is tightly bound to iron scavenging proteins, 2,2'-dipyridyl was added to the cultures to limit free iron availability in the medium (Supplementary Figure 2). Under iron limitation and in the presence of haem, or selected porphyrin intermediates of haem biosynthesis, W83 and the complementation mutant were able to grow but the growth of the ΔhusA mutant was significantly compromised (Fig. 1a). Under iron-replete conditions HusA was not expressed (Fig. 1b, Supplementary Figure 3), and the ΔhusA mutation had no effect on growth (Supplementary Figure 4). In the haem-limited condition a large proportion of HusA was found in the extracellular fraction, consistent with a haem scavenging function (Fig. 1b).

Because the gingival epithelium is the primary interface for *P. gingivalis* interaction, an epithelial cell model[11] was used to compare intracellular survival of wild type and mutant strains. Confocal fluorescence imaging revealed that *husA* deletion impaired the capacity of *P. gingivalis* to survive within epithelial cells (Fig. 1c). The number of ΔhusA mutant bacteria recovered from infected epithelial cells was reduced by ~10-fold, compared with the ΔhusA+ control and wild-type strains (Fig. 1d). Expression profiling of genes involved in haem uptake showed a significant increase (~3.5-fold) in the level of *husA* mRNA for intracellular *P. gingivalis* (Fig. 1e, Supplementary Figure 5). Taken together, these results indicate that HusA contributes significantly to *P. gingivalis* survival within epithelial cells.

**HusA has a unique structure to capture haem**. To establish the molecular basis for haem binding, the solution structure of apo-HusA was determined using NMR spectroscopy (Fig. 2a; see Supplementary Table 1 for NMR structure statistics). Purified recombinant HusA (rHusA), corresponding to the full-length mature protein (residues 24–218 without signal peptide), contains no haem ligand according to UV-visible spectroscopy. Small-angle X-ray scattering (SAXS) shows it to be predominantly a single monomeric species free of large aggregates (Supplementary Figure 6, Supplementary Table 2) as judged by: the near zero slope of logI(q) vs logq plots at low q for all samples, linear Guinier plots (Supplementary Figure 6a), and Rg and I(0) values in agreement from Guinier and P(r) analyses (Supplementary Figure 6b). The majority of NMR data were collected at two temperatures (298 K and 308 K) to assist with resonance assignment, for example to resolve overlapping signals. The final NOESY spectra used for NMR structure calculations were collected at 308 K, which gave the best compromise between signal intensity and spectral dispersion/signal completeness (temperatures in the range 278–323 K were investigated). The HusA structure comprises nine α-helices arranged as four helix-turn-helix motifs, with an additional C-terminal helix, which together form a right-handed super-helical arrangement with one concave and one convex face (Fig. 2a, in cyan).

Aside from a small number of residues in inter-helical regions, and the residues at the N- and C-termini, the HusA structure is well-ordered, as judged by NOE data and $^{15}N$ relaxation measurements (Supplementary Figure 7). Overall, the NMR structure and the shape reconstruction from SAXS data are in agreement (Fig. 2b). Dimensionless Kratky plots (Supplementary Figure 6c) have the expected bell shape for a folded protein with a small rise at qRg-values >4.5 indicating a small degree of flexibility in the structure as observed in the $^{15}N$ relaxation analysis. Structural similarity searches revealed that HusA shares topological and conformational similarity with several tetratrico-peptide repeat (TPR) proteins, including members of the 14-3-3 subclass (Supplementary Figure 8[12]). However, HusA does not conform to the strict 34-residue sequence-motif that characterizes TPR domains, nor to related 35- and 36-residue repeats. While TPR domains often function as peptide and phosphopeptide binding modules[12], HusA is an unusual example of a TPR-like fold that binds to porphyrin/haem.

During NMR structure calculations, it was noted that a small number of NOEs, in particular a subset of NOEs arising from the side chain of W130 located close to the loop region of the α5α6 helix pair, consistently remained unassigned by the automated structure calculation routines in CYANA 3[13]. Some of these NOEs appeared to have unambiguous assignments that were incompatible with the final converging structures, and instead suggested the presence of an alternative conformation of W130. Several factors supported the presence of some heterogeneity in this

region of the structure: first, the $^{1}H^{N}$–$^{15}N$ heteronuclear NOE was slightly reduced for residues in the W130 ($\alpha 5\alpha 6$) loop, and also the adjacent $\alpha 7\alpha 8$ loop (Supplementary Figure 7); second, NMR signals from the backbone of residues in the $\alpha 7\alpha 8$ loop were generally weak (A167 and Q168 resonances were weak at 298 K

and largely absent at 308 K); third, there were fewer than expected NOEs, and consequently low precision, for this region of the structure (r.m.s.d. of 1.8 Å for backbone atoms of residues 166–169 compared to 0.8 Å for residues 28–213). Excluding some of the NMR signals assigned as NOEs between W130 and

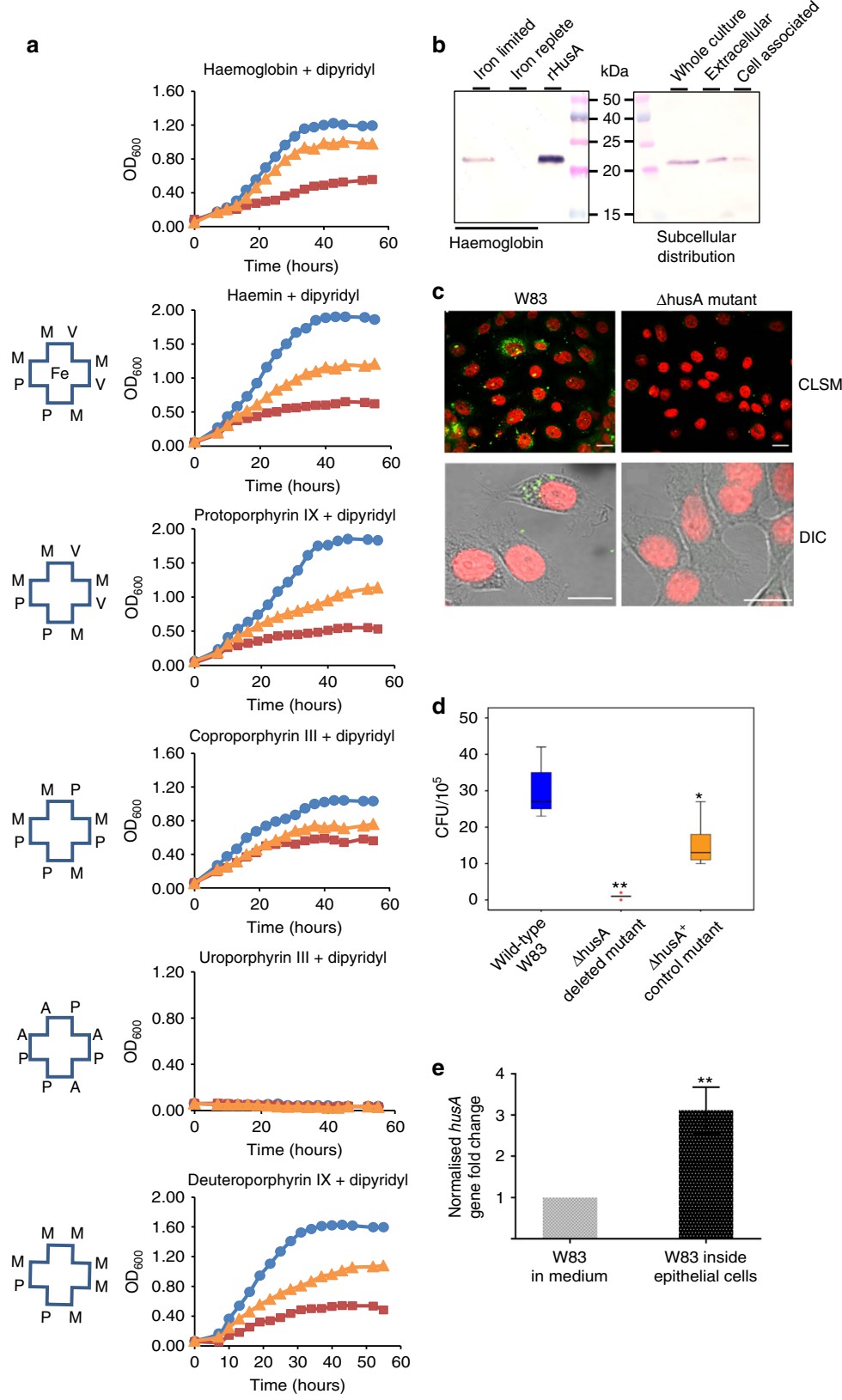

**Fig. 1** HusA is important for utilisation of haem/porphyrin and intracellular survival of *P. gingivalis*. **a** Representative planktonic growth curves of *P. gingivalis* wild-type W83 strain (blue circles), *husA* deleted mutant ΔhusA (burgundy squares), and *husA* complemented mutant ΔhusA$^+$ (orange triangles) under iron-limited conditions (100 μM iron chelator dipyridyl) supplemented with different haem/porphyrin sources, including haemoglobin (at 0.5 μM), haemin (at 2 μM), protoporphyrin-IX (PPIX; at 2 μM) and coproporphyrin III dihydrochloride (at 2 μM). Schematics for the different porphyrins used are shown with the following notations: Fe for iron, M for methyl group, V for vinyl group, P for propionate group and A for acetate group. **b** HusA protein expression and sub-cellular distribution were evaluated by immunoblotting using polyclonal anti-HusA antibodies. The bacterial biomass was adjusted to OD$_{600}$ of 0.5. **c** Representative immunofluorescence images of *P. gingivalis* invading H413 gingival epithelial cells at 24 h post infection. Confocal laser scanning microscopic (CLSM) images represent epithelial cells infected by W83 or ΔhusA mutant, respectively. Higher magnifications of DIC images represent the differential interference contrast images of field of interest. *P. gingivalis* appeared as green and the nuclei of epithelial cells appeared as red. Scale bar, 10 μm. **d** Comparison of intracellular viability of *P. gingivalis* wild type and mutants quantified by colony forming units (CFU ×10$^5$) per flask of epithelial cells. Three independent experiments with triplicate assays were conducted. Box plot shows the median value, 25th and 75th percentiles (box limits), maximum and minimum values (whiskers) and outliers (dots; **$p < 0.001$). **e** Comparison of *husA* gene expression of *P. gingivalis* in medium and within epithelial cells. Relative transcription of the *husA* gene was normalised against the housekeeping gene 16S rRNA. Results represent mean value ± standard deviation of three independent experiments

adjacent Q131 in the original conformer of W130 allowed CYANA to converge on a second ensemble of structures (Fig. 2a, in orange). Overall the two NMR ensembles of HusA had highly similar backbone and side chain conformations (with an r.m.s.d. 1.1 Å over residues 28–213), with the primary exceptions being the side chain position of W130 and the adjacent α7α8 loop (with an r.m.s.d. of 2.6 Å between the two sets of structures).

NMR titration studies were then used to map the haem binding site on HusA. Titration of haemin (Fe(III)PPIX) into $^{15}$N-labelled HusA resulted in the loss of a number of NMR signals (peak height changes are shown in Fig. 2c, black bars, based on NMR spectra shown in Supplementary Figures 9–11) but other signals remained essentially unperturbed indicating no major conformational change upon binding. The SAXS data also indicated no large-scale conformational change upon haemin binding (see Supplementary Figure 6, Supplementary Table 2). The non-paramagnetic Zn(II)-protoporphyrin-IX (ZnPPIX) titration was performed to remove potential contributions to signal loss due to paramagnetic effects from Fe(III). NMR signal intensity decreased for a very similar set of residues in the haemin and ZnPPIX titrations (Fig. 2c, Supplementary Figure 9), although some changes in the α1α2 loop and α9 were only seen with haemin. The similarity of these two data sets suggests that chemical exchange, rather than paramagnetic effects, was a major contributor to signal loss, and that haemin and ZnPPIX bound to very similar sites on HusA. The largest signal changes mapped to segments of the α3α4, α5α6 and α7α8 helix pairs that are adjacent in the 3D structure and that together form a hydrophobic groove in the concave face of HusA (Fig. 2d, dashed outline), suggesting this groove as a potential porphyrin-binding site.

To corroborate NMR titration data, selected residues from within and outside the predicted binding site were mutated and tested for haemin binding. Of the seven mutants (mutated residues are annotated in Fig. 2d) that were produced as folded proteins (Supplementary Figure 12), four mutants (L107A, R121A, Y133A, R139A) displayed similar binding to wild-type HusA according to UV-vis spectrometry (Supplementary Figure 13 and Supplementary Table 3). Mutants LV123/4NN, W130N and Y164A exhibited differences in UV-vis absorption (Fig. 2e), compared to wild-type rHusA upon haemin addition and had slightly reduced haemin affinity (by >50%) according to fluorescence quenching assays (Fig. 2f and Supplementary Table 3; individual fits and residuals are shown in Supplementary Figure 14a–c). A mutant incorporating all four mutations was folded but displayed minimal binding to haemin (Fig. 2e, Supplementary Figure 14e and Supplementary Table 3) suggesting that L123, V124, W130 and Y164 contribute in a combinatorial fashion to haem binding.

Most proteins that bind haem, including bacterial haem binding proteins characterised to date, use side chains of His, Tyr, Met or Cys to ligate the haem iron at one (or both) axial coordination site(s)[14]. Inspection of the putative haem binding site in HusA suggests that Y164 could potentially provide an axial ligand to a bound ferric haem (nearby M116 could potentially serve as a second iron ligand, in combination with Y164, but is not likely as a single haem ligand according to the haem protein database[15]). The absence of haem hyperfine shifted signals (i.e., below –2 or above 10 ppm) in $^1$H 1D NMR spectra recorded at temperatures from 298 to 328 K (Supplementary Figure 15), however, provides no evidence for iron coordination by HusA side chains. This is in contrast to the HmuY haemophore from *P. gingivalis* for which hyperfine shifted signals could be observed upon the ligation of haemin, Fe(III) mesoporphyrin IX, and Fe(III)DPIX[16,17]. To further probe a potential role for Y164 in haem binding, we constructed a Y164F mutant, which retains the phenyl ring but lacks the hydroxyl group that can coordinate ferric haem. The UV-vis absorption profile of Y164F:haemin is highly similar to HusA:haemin (Fig. 3a, compare purple and red traces) suggesting the haem binding environment in Y164F is essentially the same as for HusA wild type. The 488 nm excitation resonance Raman spectra of free haemin, HusA, or HusA Y164F in the range 1200–1700 cm$^{-1}$ showed the same peak positions (Fig. 3b), suggesting a similar 5-coordinate Fe(III) centre in all three samples. The ν10 coordination and spin-state marker band, which is strongly enhanced with Q-band excitation[18], is at 1625 cm$^{-1}$, 1626 cm$^{-1}$ and 1626 cm$^{-1}$ for haemin, HusA:haemin and HusA Y164F:haemin, respectively, typical for a 5-coordinate high-spin complex. By comparison, in the ShuT haem binding protein from *Shigella dysenteriae*, Tyr coordination of ferric haem was found to shift ν10 to a lower value of 1616 cm$^{-1}$[18]. In addition, no internal Tyr stretching modes were observed for HusA:haemin, as were evident for ShuT:haemin in Q-band excitation resonance Raman spectra. Together these data provide no evidence for a change in iron coordination when haemin binds to HusA.

The absorption band at ~603 nm in the HusA:haemin UV-vis spectrum (Fig. 3a) could be attributed to the hydroxyl oxygen of tyrosine, as seen in human serum albumin[19] and various staphylococcal haemophores[20–22], or an axial water molecule, as seen in aqueous haemin[18] or metmyoglobin[23]. UV-vis spectra recorded in the pH range from 5 to 10 showed a shift from a shoulder at ~630−640 nm to the more strongly absorbing peak at 603 nm with increasing pH (Fig. 3a, insert). This behaviour is well characterised for myoglobin, haemoglobin and free haemin, and is attributed to the deprotonation of an axial water molecule (to form hydroxide) coupled with a high-to-low spin transition in the

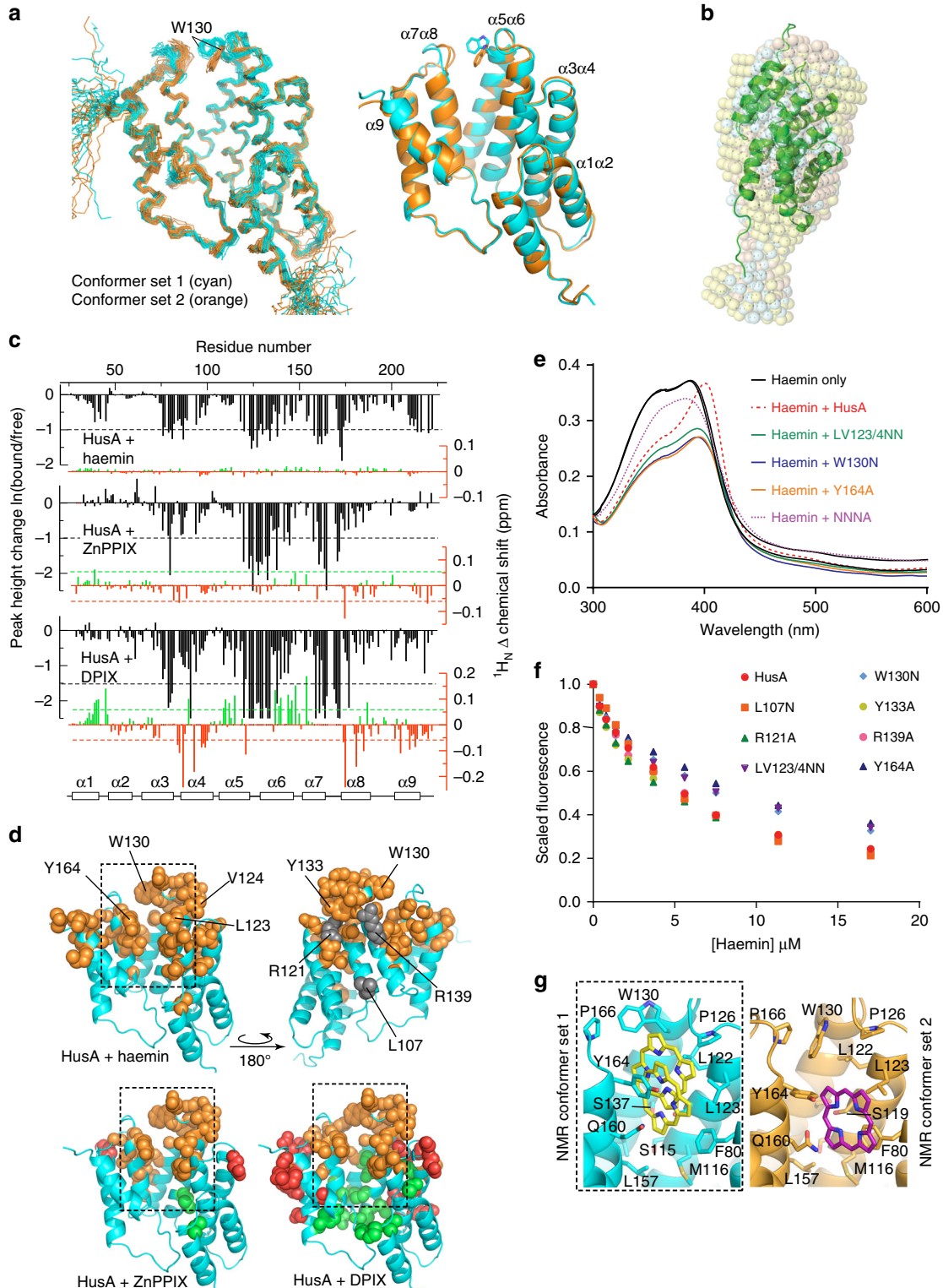

haem iron spin equilibrium[23]. The pKa of this transition for HusA and the Y164A/F mutants was in the range 6.1–6.5, similar to the pKa of 6.7 measured for aqueous haemin (Fig. 3c)[18], and similar to the pKa of 6.2 measured for murine p22HBP (haem binding protein, 22 kDa; Supplementary Figure 16) that binds porphyrins without coordinating the prosthetic metal[24]. In contrast, the pKa of the tyrosyl hydroxyl is typically >10, and, consistent with this, we saw no acid-alkaline transition for the spectra of two bacterial haemophores, IsdB and IsdH, known to ligate ferric haem through tyrosine (Supplementary Figure 16)[21,22]. Furthermore, when we mutated the haem coordinating tyrosine in IsdH (Y642A), the resulting mutant bound haem non-covalently and displayed acid-alkaline transition (pKa 5.9) which was very similar to rHusA (Supplementary Figure 16). Taken together, these results suggest that haemin binds to HusA as a five-coordinate Fe(III) complex with an axial water/hydroxide. Lack of evidence for iron coordination by a protein side chain makes HusA unusual amongst bacterial haem binding proteins,

**Fig. 2** The structure of HusA and analysis of the porphyrin/haem binding site. **a** Ensembles (stick backbone representation) of HusA conformer set 1 (cyan) and alternative conformer set 2 (orange; 20 lowest energy structures in each) calculated in CYANA 3, and the energy minimized average structures (ribbons) showing different orientations of the W130 side chain and adjacent α7α8 loop. **b** Superimposed DAMMIN models from SAXS data (semi-transparent pink, cyan and yellow spheres for HusA in reduced and standard buffers and HusA:haem, respectively) overlaid with the representative NMR structure (ribbon). **c** Mapping the porphyrin-binding site in HusA. NMR signal intensity changes (black) and $^1H_N$ chemical shift perturbations (CSPs; red/green), measured from $^{15}N$-HSQC spectra (Supplementary Figures 9–11 and Supplementary Table 5) upon addition of haemin or ZnPPIX or DPIX. Signal intensity changes are plotted for a HusA:porphyrin molar ratio of 2:1. CSPs were measured for detectable signals with porphyrin in molar excess. **d** Residues experiencing a reduction in NMR signal (orange) or up-field CSP (red) or down-field CSP (green) above thresholds shown in **c** (dashed lines) are shown as spheres. Side chains that were mutated are annotated. **e** UV-visible spectra of 10-μM haemin, alone or in the presence of 10-μM HusA or HusA mutants, recorded at pH 8.0. **f** Scaled and corrected fluorescence at 335 nm recorded from titrations of HusA or HusA mutants (1 μM concentration) with increasing concentrations of haemin. Measured fluorescence data points (symbols) were fitted to a 1:1 binding model (Supplementary Figure 14). **g** Representative pooled results for docking a porphyrin ligand (haemin, PPIX or DPIX) to the energy minimized average structure of HusA NMR conformer set 1 (cyan) or HusA NMR conformer set 2 (orange) showing the range of poses adopted (yellow, purple sticks). Only the position of the porphyrin ring, which was similar for haemin, PPIX or DPIX, is shown. Cluster analysis of all docking results is shown as Supplementary Figure 20a

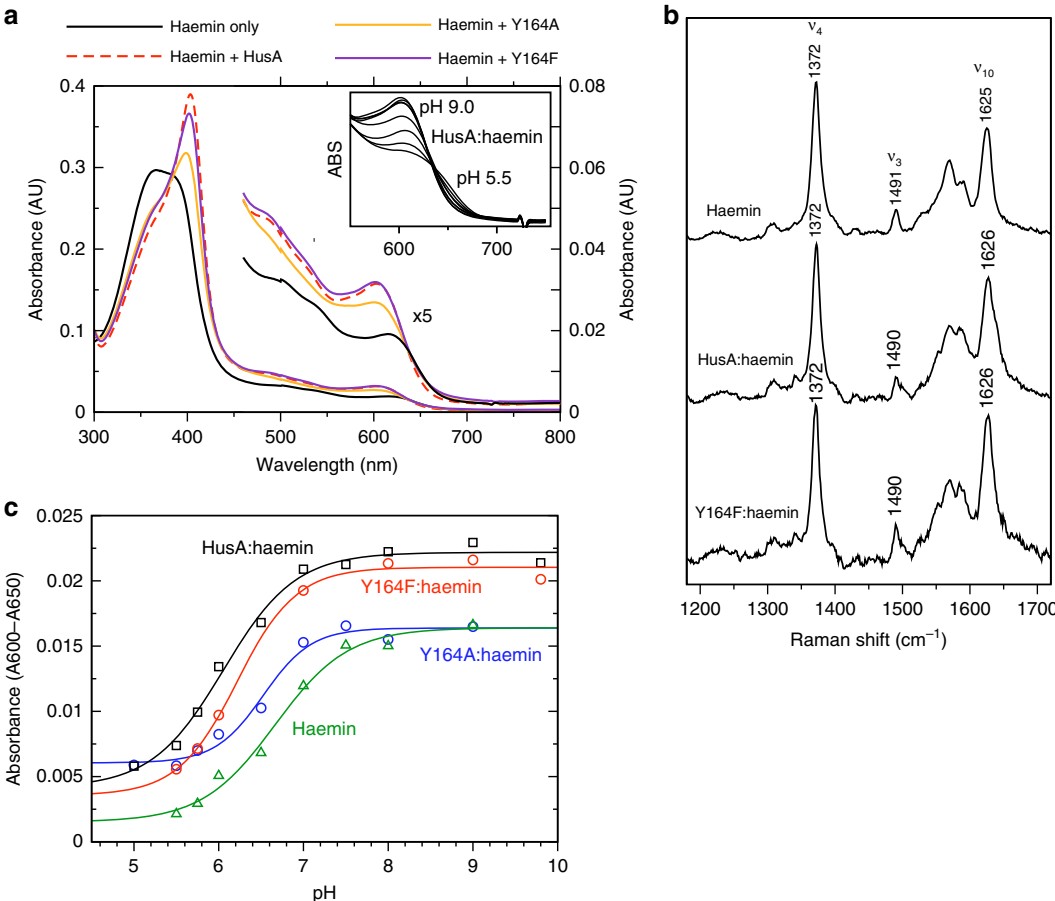

**Fig. 3** Spectroscopic studies suggest haem iron coordination is unchanged by HusA binding. **a** UV-visible spectra of 5-μM haemin in the presence of 7.5-μM HusA, Y164A and Y164F (0.2 M sodium phosphate, pH 7.0). Insert shows dependence of ~600 nm band on pH. **b** Resonance Raman spectra recorded with excitation at 488 nm for haemin (10 mM), HusA (3.2 mM) and HusA Y164F (3.5 mM). **c** Acid-alkaline transition of the absorbance band at ~600−650 nm for 5-μM haemin alone or bound with 7-μM HusA, Y164A or Y164F. Spectra were recorded in 0.1 M NaCl with 0.1 M of the following buffers: citrate, pH 5.0; BisTris-HCl, pH 5.5; BisTris-HCl, pH 5.75: BisTris-HCl, pH 6.0; sodium phosphate, pH 6.5; sodium phosphate, pH 7.0; sodium phosphate, pH 7.5; Tris-HCl, pH 8.0; sodium borate, pH 9.0; sodium borate/NaOH, pH 9.8; sodium carbonate, pH 10.0 (not all titration points were collected for each protein)

and suggests that hydrophobic interaction with the porphyrin ring is the primary binding mode, and consequently, that HusA may possess broader specificity for metallated or unmetallated porphyrins.

**HusA can bind a range of porphyrins apart from haem**. To explore ligand specificity, we investigated the ability of HusA

to bind to various porphyrin analogues. According to fluorescence quenching assays and UV-vis absorbance profiles, HusA bound to unmetallated PPIX and DPIX, as well as Fe (III)DPIX, with considerably higher affinities than to haemin, supporting the earlier conclusion that iron is not a strong determinant of HusA ligand binding (Fig. 4a–e and Supplementary Table 4; individual fits and residuals are shown in Supplementary Figure 14d). NMR titration with DPIX

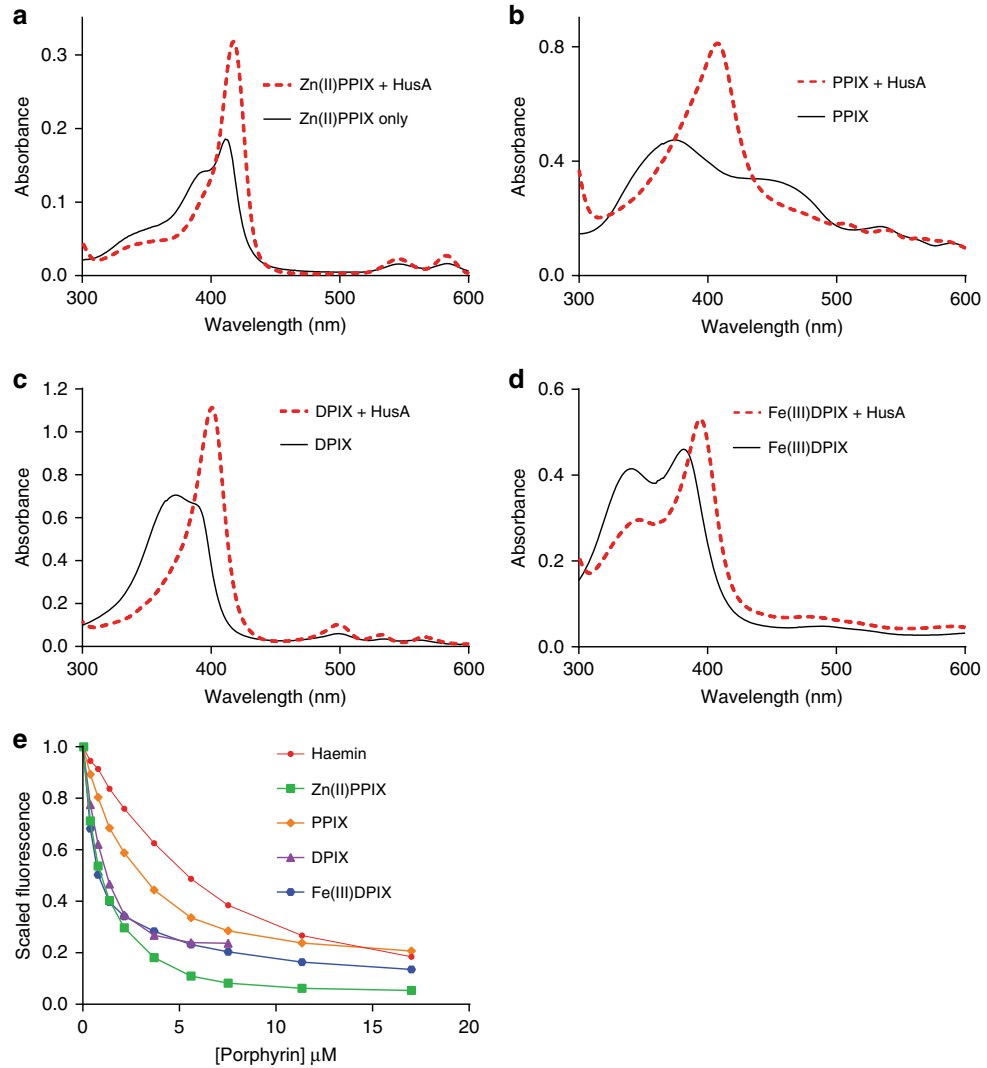

**Fig. 4** HusA binds a range of porphyrins, including metallated and unmetallated forms. **a–d** UV-visible spectra of 10−20 μM of ZnPPIX, PPIX, DPIX and haemin, alone and in the presence of 1–1.5 molar equivalents of HusA. **e** Scaled and corrected fluorescence intensities at 335 nm during titration of 1 μM of HusA with increasing concentrations of haemin and other porphyrins. Fitting to a 1:1 binding model yielded the dissociation equilibrium constants for haemin (7.3 ± 1.1 μM), PPIX (1.9 ± 0.6 μM), ZnPPIX (1.3 ± 0.4 μM), DPIX (0.36 ± 0.09 μM) and Fe(III)DPIX (0.22 ± 0.05 μM), in order of increasing affinity

resulted in loss of a similar set of NMR signals as seen during haemin and ZnPPIX titrations (Fig. 2c, black bars; direct comparison of HSQC spectra is shown in Supplementary Figure 18), suggesting that the DPIX-binding site on HusA is the same as for haemin. In addition to loss of signal intensity, some residues exhibited $^1H^N$ chemical shift perturbations (CSPs; Fig. 2c, red/green bars) in the intermediate-to-fast exchange regime (Supplementary Figures 10, 11). These CSP effects mapped to residues slightly more distant from the putative porphyrin/haem binding site (Fig. 2d, red and green spheres), whereas residues closer to the binding site remained in intermediate exchange (signals from these residues declined in intensity, and were not recovered even at 2–3 molar excess of ligand). In contrast to porphyrins, the linear tetrapyrroles, biliverdin and bilirubin, showed no interaction with HusA as detected by UV-visible absorption spectral analysis (Supplementary Figure 19a, b). Together, these results suggest that the porphyrin ring structure is a major determinant of binding specificity. Similarly, HusA did not appear to bind the haem biosynthesis intermediates coproporphyrin III and

uroporphyrin III (Supplementary Figure 19c, d), possibly due to their greater number of ionisable carboxyl groups.

Unbiased docking studies using the HusA solution structure independently identified a single binding groove for haemin, PPIX, and DPIX that agreed well with NMR titration and mutagenesis data (Fig. 2g, Supplementary Figure 20a). The docking site was bounded by the side chains of F80 (α3α4), M116, L122, L123, P126, W130 (α5α6), and Y164, Q160 and L157 (α7α8), which are mostly hydrophobic in character (Fig. 2g). Within this groove the porphyrins could adopt a variety of orientations due to rotation in the plane of the porphyrin, resulting in different positions/interactions of the propionate side chains (to emphasise the conserved interactions with the porphyrin ring, only the position of the porphyrin skeleton, without various side groups, is shown in Fig. 2g). Interestingly, the docked orientation of the porphyrin plane was consistent with the pattern of CSPs obtained from DPIX NMR titrations, according to the prediction of up-field shifts for residues above/ below the porphyrin plane and down-field shifts for residues at the periphery of, or in-plane with, the porphyrin, due to strong

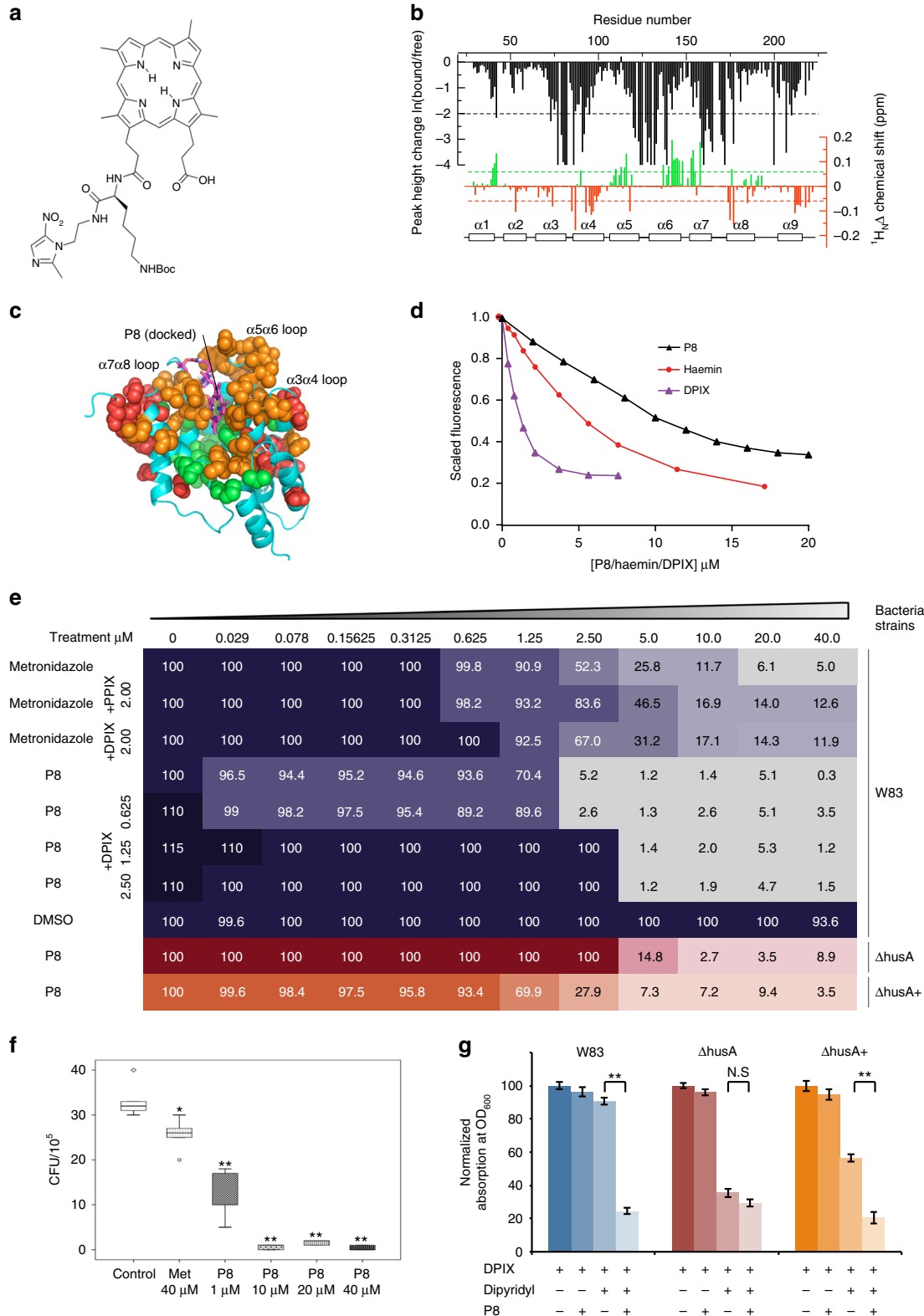

ring current effects. Similar docking solutions were obtained for both NMR conformer sets and for all three ligands (analysis shown in Supplementary Figure 20a); although, the 'inward' facing position of the W130 indole ring (NMR conformer set 2) tended to restrict binding to one end of the hydrophobic groove (Fig. 2g, purple sticks). Docking solutions with inward facing W130 side chain were also obtained when docking to NMR

conformer set 1, due to reorientation of side chains around the binding site as part of the docking algorithm (Supplementary Figure 20a). Docked ligands consistently adopted π-stacking interactions with the benzene ring of F80 on one side of the porphyrin ring, and/or π-stacking or edge-to-face stacking with the phenol ring of Y164 on the opposite side of the porphyrin (Fig. 2g). A cluster of Ser115, Ser119, Ser137 and Ser161 residues

**Fig. 5** A prototype DPIX-conjugated antibiotic binds HusA and inhibits *P. gingivalis*. **a** The chemical structure of the deuteroporphyrin-lysine-metronidazole adduct, P8. **b** NMR signal intensity changes (black) and $^1H^N$ CSPs (red/green) for HusA upon addition of P8 (measured from $^{15}N$-HSQC spectra shown in Supplementary Figure 21). **c** In silico docking of P8 to the energy minimized structure of HusA (conformer set 1), with residues experiencing NMR signal loss (orange) or CSPs (red/green) mapped to the structure (colouring as described in Fig. 2d). **d** Scaled and corrected fluorescence at 335 nm during titration of 1 μM of HusA with increasing concentrations of P8, haemin or DPIX. **e** Comparison of P8 inhibitory effect towards *P. gingivalis* wild-type W83 and *husA* mutants using checkerboard analysis. Serial dilutions of antibiotics (horizontal axis) were applied to a 96-well plate with different *P. gingivalis* strains (vertical axis). Serial dilutions of DPIX were also supplemented into wild-type W83 group to evaluate the competitive effect against P8. Absorption at $OD_{600}$ of each well was recorded after 24 h of incubation. The $OD_{600}$ of each strain grown in the absence of antibiotics was defined as 100% growth, which was applied to normalise the reading collected from other wells. The normalised percentage of each well was plotted in the checkerboard. Metronidazole alone and in combination with PPIX or DPIX was applied as control. **f** Intracellular killing assay of P8 antibiotics. Gingival H413 epithelial cells were infected by *P. gingivalis* W83 and then treated with P8 at various concentrations. Recovered bacteria were determined by CFU count. Metronidazole and DMSO were used as control. Three independent experiments with triplicate assays were conducted. Box plot shows the median value, 25th and 75th percentiles (box limits), maximum and minimum values (whiskers) and outliers (dots; \*\**p* < 0.001). **g** Iron restriction induced HusA expression and enhanced P8 growth inhibition of *P. gingivalis* in the presence of 100-μM dipyridyl. Wild-type W83 was susceptible to P8 at 1 μM, a concentration below the MIC; error bars, standard deviations

located at the base of the binding groove provided hydrogen bond donors to one or both propionates in 70% of docking solutions; however, the empirical scoring function used did not compute desolvation energies and so the docking studies shown here cannot determine whether buried or solvent exposed propionates would be energetically more favourable. In summary, the docking analysis identifies a hydrophobic binding groove that is large enough to dock porphyrin in a number of poses, although, we note that protein side chain reorganisation, particularly at the interface between α5α6 and α7α8 could generate different local interactions from those shown in Fig. 2g. In addition, the absence of suitably placed basic groups that could provide strong orientation preference for the bound porphyrin, together with the absence of NMR signals from ligand-site residues even at saturating porphyrin concentrations (relative to the measured $K_d$), raise the possibility that ligands might exist in a range of bound sub-states; indeed, this mode of binding could be a mechanism to achieve the broad substrate specificity observed for HusA.

**A prototype DPIX-antibiotic conjugate targets HusA.** We previously designed and synthesized porphyrin adducts with restricted antimicrobial specificity against *P. gingivalis*[25,26]. Docking indicated that a prototype antibiotic, a DPIX-lysine-metronidazole adduct (termed P8; corresponding to Adduct 12[25]; Fig. 5a) could be accommodated in the haemin binding site, with the porphyrin moiety adopting a similar set of protein side chain interactions as seen for the free porphyrins (Fig. 5b, c). NMR titration analysis (Fig. 5b) showed signal loss for residues close to the proposed binding site, in a pattern similar to that seen for DPIX (direct comparison of the HSQC spectra is shown in Supplementary Figure 21a), as well as similar patterns of up-field and down-field shifts as seen for DPIX (Supplementary Figure 21b). These patterns are consistent with the porphyrin rings of P8 and DPIX ligands adopting similar orientations when bound to HusA. Taken together, the data suggest that P8 binds at a similar site to haemin and DPIX. P8 bound HusA according to UV-visible absorption and fluorescence data with an affinity in the 1–10 μM range (Supplementary Figure 22a and Fig. 5d).

The inhibitory effect of P8 against planktonic *P. gingivalis* W83 was determined by a chequerboard assay. The minimal bactericidal concentration (10–20 μM) was determined as the lowest concentration of P8 where no growth recovery was recorded on the eTSB agar plate. The minimal inhibitory concentration of P8 (1.25–2.5 μM) is ~fourfold lower than that of metronidazole controls (5–10 μM) (Fig. 5e). The inhibitory effect of P8 declined with increasing density of bacteria exposed (Supplementary Figure 22). A reduced efficacy of antibiotics with

increased bacterial density has been reported previously[27]. P8 was more effective than metronidazole in killing extracellular and intracellular *P. gingivalis* (Fig. 5e, f) and this correlated with progressive accumulation of P8 over a 2-h period (Supplementary Figure 23b, c). Neither PPIX nor DPIX sensitised *P. gingivalis* cells towards metronidazole under either iron replete or iron-limited conditions (Supplementary Figure 24). The mediator role of HusA in the bactericidal action of P8 was supported by the observation that expression of HusA sensitizes wild-type *P. gingivalis* to P8 concentrations lower than the P8 minimum inhibitory concentration, under iron-limited conditions when HusA is highly upregulated (Fig. 5g).

## Discussion

Despite the presence of multiple haem uptake systems in *P. gingivalis*, our work demonstrates a key dependency for survival and growth of the bacteria on expression of HusA under physiological conditions where free iron is scarce. Similar to *P. gingivalis* HmuY haemophore[28,29], HusA is detected attached on the outer membrane and also released into the extracellular milieu under iron-limited conditions. Disabling the *husA* gene significantly impairs the capacity of *P. gingivalis* to survive within epithelial cells. While the data from in vitro haemin titration experiments fit well to a 1:1 binding model, the hydrophobic binding groove on HusA could potentially accommodate dimeric haem that is present in *P. gingivalis* pigment extract[10]. We found that *P. gingivalis* outer membrane extract, but not rHusA, demonstrated reverse ferrochelatase activity, whereby iron is removed from haem (Supplementary Figure 17). Previous studies suggested that a potential chelatase, IhtB, on the outer membrane surface could facilitate this process to generate metal-free porphyrins that would subsequently be taken up by *P. gingivalis*[28,30]. This extracellular metal free porphyrin could be the preferred ligand for HusA, with uptake mediated by the TonB-dependent receptor HusB, thus providing a mechanism to satisfy the porphyrin requirement of *P. gingivalis* in a biofilm. The ability of HusA to bind a wide range of porphyrins including haem precursors may allow *P. gingivalis* to successfully colonize and invade host tissues and enable intracellular survival in vivo (Supplementary Figure 25). We found that coproporphyrin III (but not uroporphyrin III) could support *P. gingivalis* growth (Fig. 1a and Supplementary Figure 3), consistent with a previous report that *P. gingivalis* lacks the enzyme to convert uroporphyrinogen III into coproporphyrinogen III but retains the enzymes that convert coproporphyrinogen III into PPIX and then to haem[31].

HusA consists of a helical bundle with a hydrophobic groove that binds haem in the absence of obvious haem iron coordination; this is very unusual amongst bacterial haem scavenging

proteins. Other bacterial haemophores typically coordinate the haem iron through one or two amino acid side chains, such as Tyr[20], Tyr/Met[22], Tyr/His[32], His/His[16] or Met/His[33] and, in these cases, mutation of the haem ligating residues typically results in over 400-fold reduction in haem affinity[34–36], suggesting high specificity for the metallated porphyrin. In contrast, HusA utilises a more generalist porphyrin-binding mode that relies only on hydrophobic and π-stacking interactions with the porphyrin. Y164 is the only residue within the docking site with the potential to coordinate the haem iron, and mutation of this residue causes only a modest twofold loss of binding affinity. In this regard, HusA resembles the mammalian intracellular haem transporter p22HBP which lacks iron-ligating residues and binds haem and a range of non-metallated haem synthesis intermediates with affinities in the range of 0.5–25 nM[24]. While we have previously reported the HusA:haemin binding to be of high affinity ($K_D$ ~2 nM)[10], our current investigations using a range of methodologies suggest that the binding affinity is far weaker and is in the low micromolar range. A micromolar binding affinity is more consistent with the intermediate exchange observed in the NMR titration experiment, and the loss of bound ligands during chromatographic separations. The importance of weak and transient interactions has been increasingly recognized in biology[37] and in the case of HusA, this likely facilitates transfer of the acquired porphyrin to its cognate outer member receptor, HusB. The NMR structure reveals that HusA has structural similarities with TPR family proteins. Interestingly, TPR proteins have acquired a wide variety of roles in bacterial pathogenesis[23]. HusA is, however, currently the only example of a TPR-like fold with a porphyrin/haem binding function, suggesting that the TPR-like fold has been co-opted to porphyrin/haem binding during the evolution of *Porphyromonas* species. We propose that the large ligand binding groove of HusA and the demonstrated broad specificity for porphyrin ligands are properties that can be exploited to gain further improvements of prototype porphyrin antibiotics that target *P. gingivalis*.

The prototype antibiotic P8, a deuteroporphyrin-metronidazole conjugate, demonstrated enhanced inhibition of *P. gingivalis* growth under conditions where HusA is highly expressed. We propose the bactericidal action of P8 to be mediated by reduction of the imidazole nitro group of attached metronidazole[38] following uptake by *P. gingivalis* mediated primarily by HusA. The other haem uptake systems of *P. gingivalis* including HmuY, may also contribute to P8 uptake although lack of iron coordination would significantly compromise the binding affinity of P8 for HmuY[39]. Enhanced intracellular killing by P8 compared with metronidazole can be attributed to both facilitated passage into the epithelial cell and increased expression of HusA by intracellular *P. gingivalis*.

Finally, the general strategy of targeting nutrient scavenger proteins by strategic arming of substrates[21] critical for successful colonization or invasion of the host may be applicable for selective eradication of pathogens while minimizing toxic effects on commensal bacteria and thus reducing the development of antibiotic resistance.

## Methods

**Bacterial strains and cultures and cell lines**. *P. gingivalis* wild-type W83 and mutant derivatives were grown in enriched Tryptic Soy Broth (eTSB) supplemented with 0.5 g L$^{-1}$ of L-cysteine, 2 mg L$^{-1}$ of menadione, and haemin or haemoglobin where indicated at 37 °C in an anaerobic chamber (Don Whitley Scientific Limited, UK) with an atmosphere of 80% N$_2$, 10% CO$_2$ and 10% H$_2$. *Escherichia coli* strains DH5α (Invitrogen) and Rosetta™ DE3 (Novagen) were used for plasmid construction and recombinant protein expression. *E. coli* was grown at 37 °C in lysogeny broth (LB) broth and agar supplemented with ampicillin (100 μg mL$^{-1}$) and kanamycin (50 μg mL$^{-1}$), where necessary.

The oral epithelial cell line H413 displaying stratified epithelial cell morphology was grown in Joklik's minimal essential medium (JMEM, Sigma-Aldrich) supplemented with fetal calf serum (0.5–10%, where indicated) at 37 °C in 5% CO$_2$ atmosphere.

**Site-directed mutagenesis**. The primers used for construction of mutants are listed in Supplementary Table 6. Expression plasmids of various site-directed mutagenic recombinant proteins were constructed using a modified version of the site-directed, ligase-independent mutagenesis method described previously by Chiu et al.[40]. Briefly, a set of four primers with an 18-base complementary overhang (Supplementary Table 6) was applied to amplify the whole-plasmid starting from the region to be mutated using AccuPrime Pfx DNA polymerase (Invitrogen). The amplified PCR products were digested by DpnI, and then denatured at 95 °C for 1 min, followed by hybridisation using two cycles of 68 °C for 5 min, 60 °C for 5 min, 55 °C for 5 min, 30 °C for 10 min. The hybridised product was transformed into *E. coli* DH5a cells using a modified version of the Hanahan method. Clones were selected on antibiotic selective LB agar and screened for the correct constructs by DNA sequencing of the pertinent region. The verified plasmids were then transformed into the protein expression host *E.coli* Rosetta™ DE3 for recombinant protein production.

**Growth curve studies**. Growth curve experiments were conducted as described previously by Gao et al.[10]. Briefly, fresh colonies of *P. gingivalis* wild-type W83, ΔhusA and ΔhusA$^+$ mutants were inoculated into eTSB media without haem supplementation as the starter cultures. Haem stores were depleted by daily passage of the parental cultures into haem free media at 1:10 inoculum until the biomass of the following passage could no longer attain an OD$_{600}$ of 0.5 after 24 h of incubation. The penultimate culture was adjusted to OD$_{600}$ of 0.5 and a 1:10 inoculum was transferred anaerobically into 5 mL of eTSB with various supplementations (combinations of 100 μM dipyridyl, various haem and porphyrin sources) in individual screw cap tubes (Sarstedt, Australia). The experiment was set up and conducted in an anaerobic chamber (Don Whitley Scientific Limited, UK). As the iron normally present in host tissues is tightly bound to iron scavenging proteins, dipyridyl was added to the cultures to remove free iron to mimic the growth conditions in vivo[41]. Notably, the trypticase soy broth (TSB) contains 0.74 μg mL$^{-1}$ iron determined by atomic absorption spectrophotometry[42]. Iron derived TSB requires 500 μM dipyridyl[43]. At pre-determined time points, tubes were vortexed and absorbance at 600 nm was recorded using a DiluPhotometer™ Spectrophotometer (Implen Inc., Germany). Three independent experiments were performed in triplicate.

**Infection of epithelial cells with *P. gingivalis***. Human H413 gingival epithelial cells were cultured to confluence (90–95%) for infection. Haem-depleted *P. gingivalis* cells, including wild-type W83, ΔhusA and ΔhusA$^+$ mutants were collected and washed twice with cold sterile phosphate-buffered saline (PBS). Bacterial cells resuspended in JMEM with 0.5% FCS were applied to challenge H413 cells at a multiplicity of infection (MOI) of 100 and co-incubated at 37 °C in 5% CO$_2$ for 16 h. External, non-adherent bacteria were removed by washing three times in PBS and external adherent bacteria were killed by incubating with 300 μg mL$^{-1}$ of gentamicin and 200 μg mL$^{-1}$ of metronidazole for 2 h[11]. Infected cells were then washed three times with PBS, and internalised bacteria were recovered by lysing infected cells in sterile cold Milli-Q water for 20 min[11]. Dilutions of lysates were plated on blood agar plates supplemented with haemin and menadione and cultured anaerobically. Colony-forming units (CFU) of invasive organisms were then enumerated. The results reported are from three independent experiments.

**Cell fractionation and western blotting**. Bacterial cultures were adjusted to an OD of 0.5 at 600 nm and treated with 1% (v/v) protease inhibitor cocktail (Sigma-Aldrich) and 4 mM Nα-tosyl-lysine chloromethyl ketone (TLCK) (Sigma-Aldrich) for 2 h at room temperature. Bacteria were pelleted by centrifugation at 6000 × g for 30 min and the supernatant was further ultracentrifuged at 150,000 × g for 1 h to separate outer membrane vesicles and extracellular soluble fraction. Equal amount of samples were separated by SDS-PAGE and electroblotted onto 0.2 μm pore-size nitrocellulose membranes (Bio-Rad). Rabbit anti-HusA polyclonal antibody produced through a subcontractor (Genscript Inc.) was applied as primary antibody to probe the membrane at 1:5000 dilution in TBST buffer at 4 °C overnight. Alkaline phosphatase-conjugated goat anti-rabbit IgG (Dako Corp.) was used as the secondary antibody.

**Fluorescence microscopy assay**. Human gingival epithelial cells were grown to confluence and seeded on plastic chamber slides (75 × 25 mm$^2$) for incubation in 5% CO$_2$ atmosphere overnight. Bacterial cells were co-incubated with epithelial cells at 37 °C for 2 h. After exposure to the antibiotics, epithelial cells were washed in PBS and incubated in culture media containing 5% FCS for 24 h. Subsequently, samples in the chamber slides were fixed with 4% fresh paraformaldehyde in PBS for 30 min at room temperature, washed in PBS, and permeabilised by ice-cold methanol for 5 min. Samples were then blocked overnight with 10% FCS supplemented with 0.05% (v/v) Triton X-100 in PBS buffer. Intracellular *P. gingivalis* was stained using the primary antibody IIB2[44] and fluorochrome-conjugated secondary

antibody. Epithelial cells were revealed by ProLong Gold with DAPI (Invitrogen). Labelled epithelial cells with invaded bacteria were detected using an Olympus FV1000 confocal laser scanning microscope. In addition, the intrinsic fluorescence of P8 allowed temporal tracking of internalisation of P8 into epithelial cells in a separate experiment. Live cell images were recorded using a DeltaVision Elite Core Microscope.

**RNA extraction and quantitative RT-PCR.** Total RNA from planktonic bacteria or bacteria co-incubated with epithelial cells was extracted using Trizol (Invitrogen), according to the manufacturer's instructions. Complementary DNA (cDNA) was randomly primed using the AffinityScript qPCR cDNA synthesis kit (Agilent Genomics, Australia) as per the manufacturer's instruction. Detection of gene expression was subsequently performed in triplicate with a 1:5 dilution of cDNA using TaqMan probe-based real-time PCR assay on a Stratagene Mx3005P[TM] system (Agilent Genomics, Australia). The primers and probes are listed in Supplementary Table 7. All quantification data were normalized against 16S rRNA. Normalization and calibration methods were the same as in our previous publication[10].

**Protein purification for HusA and mutants.** A fragment of the HusA gene encoding the mature protein after cleavage of the N-terminal signal peptide (residues 24–218) was cloned and expressed with a C-terminal hexa-His tag in *E. coli* strain BL21(DE3) Rossetta, as previously described[10]. For more details about the purification process and the production of labelled proteins, Supplementary Methods. All experiments using purified HusA/mutants were performed in a standard buffer (50 mM Tris-HCl, 150 mM NaCl, pH 8), unless otherwise stated.

**Preparation of porphyrin stocks.** Except for PPIX, 10 mM porphyrin stock solutions were prepared freshly before use by dissolving porphyrin powder in 0.1 M NaOH, pH 12, by gentle pipetting and inversion for 5 min. Samples were then incubated on the bench for 10 min before use. The 10 mM stock solution of PPIX was prepared by dissolving PPIX powder in pure DMSO in a similar manner. Other preparation methods used in specific experiments are described in the relevant sections.

**NMR spectroscopy and NMR structure calculations.** All NMR spectra were acquired on Bruker Avance III 600 or 800 MHz spectrometers equipped with 5-mm triple resonance TCI cryogenic probeheads (Bruker, Karlsruhe, Germany), processed using Topspin 2.1 or 3.2 (Bruker Biospin Ltd) and analysed with Sparky (T. D. Goddard and D. G. Kneller, University of California, San Francisco). [1]H spectra were referenced to sodium 2,2-dimethyl-2-silapentane-5-sulfonate (DSS) at 0 ppm. HusA/mutant proteins were diluted with standard buffer to a final concentration of 50–100 μM and $D_2O$ and sodium 2,2-dimethyl-2-silapentane-5-sulfonate (DSS) were added to a final concentration of 5% (v/v) and ~20 μM, respectively. Samples were placed into 5-mm Shigemi tubes (SHIGEMI Co. Japan) and [1]H spectra were recorded with the standard Bruker pulse program p3919gp. Purified isotopically labelled HusA samples in 10 mM sodium phosphate buffer, pH 6.9, without reducing agent, were concentrated to 0.3–1.5 mM ($\varepsilon_{280}$ = 31,400 $M^{-1}$ $cm^{-1}$ calculated from the protein composition). To assist with assignment, data were collected at 298 K and 308 K and [15]N and [13]C chemical shifts were referenced indirectly using DSS according to magnetogyric ratio. NOESY spectra for structure determination were recorded at 308 K. NOEs were calibrated and assigned in an automated fashion using the NOEASSIGN macro within CYANA 3[13]. Final structure calculations in a shell of explicit water molecules were performed using XPLOR-NIH version 2.45[45].

**[15]N-HSQC titrations.** Haemin (Frontier Biosciences) at a stock concentration of 5 mM in 0.1 M NaOH was titrated into [$U$-[15]N]rHusA at 0.4 mM in 0.1 M sodium phosphate buffer, pH 6.8. ZnPPIX was prepared using PPIX (Frontier Biosciences) as the starting material. PPIX (0.5 g) was dissolved in boiling chloroform (100 mL), to which a saturated solution of Zn acetate in MeOH (1 mL) was added. The mixture was refluxed for 20 min and then a small amount of MeOH was added and, after cooling, the dark red solid was filtered off. ZnPPIX was redissolved in 80% MeOH, 20% $H_2O$, 0.01 M ammonium acetate, pH 5.2 and adjusted to ≤40% MeOH by evaporation. The final ZnPPIX preparation (1.2 mM; $\varepsilon_{556}$ = 13,500 $M^{-1}$ $cm^{-1}$ as PPIX in 2 M HCl) was >85% pure and haemin/PPIX contamination was <5% by HPLC. ZnPPIX was titrated into [$U$-[15]N]rHusA at 0.08 mM concentration in 0.1 M sodium phosphate buffer, pH 6.8. A matched buffer sample was prepared in parallel in order to control for solvent/buffer effects during the NMR titration. For each titration point, [15]N-HSQC spectra were acquired and processed with the same parameters using TOPSPIN 3. The signal-to-noise (SN) of each resolved amide signal was measured in SPARKY. The effect of ligand binding on the peak height at each residue position was calculated as $\log_e(SN_{HusA\_titration}/SN_{HusA\_alone})$, where the HusA titration condition was either HusA with ligand, or matched buffer control. The threshold for residues undergoing a substantial change in peak height was set at approximately half the maximum peak height change in the spectrum, as shown by the black dashed lines in Figs. 2c and 4b.

**Molecular docking.** Docking was performed against the HusA conformer set 1 and 2 average structures obtained after water refinement. Equivalent results were obtained using individual members of the NMR ensemble. Ligand file formatting and protonation states were adjusted using OPEN BABEL[46], and AUTODOCK TOOLS (http://autodock.scripps.edu/) was used to assign Gasteiger charges and set the number of rotatable bonds for input into the docking program VINA[47]. For details about the conformations used for the different porphyrins and the docking procedures, see Supplementary Methods.

**UV-visible spectroscopy.** All absorbance spectra were recorded at room temperature using 10-mm quartz cuvettes on a Shimadzu UV-1601 or Jasco V-630 spectrophotometer. Mixtures of HusA/mutant with porphyrins were prepared by dissolving porphyrin stocks in standard buffer to a final concentration of 5–20 μM with a 1–1.5 molar ratio of protein to porphyrin, as specified in the relevant figure legends. PPIX:HusA mixtures were prepared by incremental addition of 2–4 μM HusA with a 10–15 min wait after each addition of HusA.

**Resonance Raman spectroscopy.** Samples were analysed using a Renishaw Raman InVia Reflex microscope (Renishaw plc., Wotton-under-Edge, UK), equipped with an air-cooled charge-coupled device (CCD). The spectrometer was fitted with a holographic notch filter and a 2400 mm per line grating. The attached microscope was a Leica DMLM. The spectrometer was controlled by PC using the instrument control software (Renishaw WiRE™, Version 4.4). Before data collection the instrument was calibrated using a silicon internal standard. Sample excitation was achieved using an Argon laser (Renishaw plc., Wotton-under-Edge, UK) emitting at 488 nm. Spectra were recorded using the ×20/0.40 NA objective over the spectral range of 1200–1750 $cm^{-1}$ with the accumulation of 200 scans and 2–5 s exposure at a laser power of 100%. Spectra were not corrected for instrument response. Spectra were baseline corrected using a linear function with Spectra-Gryph1.2 (https://www.effemm2.de/spectragryph/about.html).

**Tryptophan fluorescence quenching assays.** The quenching of intrinsic tryptophan fluorescence of HusA upon porphyrin additions was measured using a Carey Eclipse Fluorescence Spectrophotometer (Agilent) set to a photo multiplier effect (PME) of 850. A 5-mm path length quartz cuvette was used and all spectra were acquired at room temperature. Excitation was set to 295 nm and emission was measured from 300 nm to 500 nm using excitation/emission slit widths of 5 nm. Binding assays were performed at a protein concentration of 1 μM HusA in standard buffer at a working volume of 0.5 mL. Fluorescence intensities were recorded 10 min after each porphyrin titration until a final porphyrin concentration of 7 μM was reached for DPIX, and 17 μM was reached for haemin, ZnPPIX and PPIX. The fluorescence quenching of HusA by PPIX was investigated in a similar manner with the following difference: the fluorescence intensities of ten 1 μM HusA samples were measured each with increasing amounts of PPIX after a 2-h incubation period to allow equilibrium to be reached. See Supplementary Methods for correction for inner filter effect and curve fitting.

**Small angle X-ray scattering SAXS.** The data were measured at the Australian Synchrotron on the SAXS/WAXS beamline[48]. For more complete information[49], see Supplementary Methods. In brief, all three forms of HusA were assessed as greater than ~95% monomeric, have similar values for Rg and dmax, and similar P (r) profiles indicating that there are no large-scale conformational changes or pronounced differences in oligomerisation states between the three forms. Using default parameters and P1 symmetry, 20 independent ab initio shape calculations were undertaken using DAMMIF for the averaged apo-HusA in reduced and standard buffers and for HusA:haem in standard buffer, yielding a set of similar shapes as indicated by normalised spatial discrepancy (NSD) values <1. We therefore ran the finer grained DAMMIN ab initio modelling in the slow mode and found very similar shapes for all forms, consisting of a globular lobe with dimensions that fit the NMR structure and a small tail extending in one direction that is characteristic of samples containing flexibility at the N- and C-termini (Fig. 2b). Alternatively, it may arise from a small amount of aggregate. In either case, more detailed atomistic modelling is not advised.

**Cytotoxicity assay.** As P8 could penetrate epithelial cells, the cell membrane integrity was tested using lactate dehydrogenase (LDH) assay. LDH is readily released following cellular membrane damage. As per manufacturer's instructions (Thermo Fisher Scientific, Australia), H413 cells were treated with P8 at various concentrations and incubated for 1 h and 2 h, respectively. The drug solubilisation medium DMSO and metronidazole were used as controls. The positive control provided by the manufacturer was used to determine the maximal LDH release and the ratio of corrected values obtained for the experimental groups relative to maximum LDH release represented cytotoxicity. The fluorescence intensity was recorded on SpectraMax® i3 (Molecular Devices, Australia).

**P8 stability assay.** The stability of P8 in the presence of epithelial cell lysate was evaluated using a fluorometric assay. Briefly, ~10[6] H413 the cells were collected by scraping and lysed by a brief sonication in pulsed mode at a power setting of 17 W

on ice. A serial dilution of P8 (5–80 μM) was incubated with cell lysate in PBS in a 96-well plate. The control was P8 incubated in PBS buffer only. The relative fluorescence (RF) was acquired every 15 min for 24 h at 37 °C in SpectraMax® i3 (Molecular Devices, Australia), using excitation at 380 nm and detection at 610 nm. The normalised relative fluorescence was calculated by the equation $RF_{normalised} = (RF_{test}−RF_{control})/RF_{control} \times 100\%$. The experiment was performed in triplicate and the values presented are the average.

**P8 antimicrobial potentiation test**. The checkerboard assay was conducted according to CLSI guidelines. The minimum inhibitory concentration (MIC) and minimum bactericidal concentration (MBC) of P8 were determined by a broth micro-dilution and CFU counting assay. Briefly, 24-h cultures of *P. gingivalis* were diluted in fresh eTSB without haemin to obtain an $OD_{600}$ of 0.2. Equal volumes (100 μL) of bacteria and P8 serial dilution (from 500 μg mL$^{-1}$; 1:2 dilutions) in culture medium were mixed into the wells of a 96-well microplate. Control wells with bacteria or P8 only were also prepared, while metronidazole was used as a reference antibiotic and DMSO as a reference solvent. After an incubation of 24 h at 37 °C under anaerobic conditions, bacterial growth was recorded at $OD_{600}$ using SpectraMax i3. The MIC values were determined as the lowest concentrations at which no apparent bacterial growth occurred. To determine the MBC values, aliquots (5 μL) of each well showing no visible growth were spread on eTSB blood agar plates and incubated anaerobically for 4 days at 37 °C. The MBC values were determined as the lowest concentrations at which no colony formation occurred. All assays were performed in triplicate.

The effect of HusA on P8 killing was further evaluated using a similar method to the one described above. *P. gingivalis* wild-type W83, ΔhusA and ΔhusA$^+$ mutants were daily passaged in eTSB media without haem supplementation to deplete haem stores. Bacteria were collected and adjusted to $OD_{600}$ of 0.2 and subsequently incubated in eTSB supplemented with DPIX or P8 only, DPIX plus dipyridyl, or DPIX plus dipyridyl and P8. The cultures were incubated anaerobically at 37 °C. At pre-determined time points, tubes were vortexed and absorbance at 600 nm was recorded using a DiluPhotometer$^{TM}$ Spectrophotometer (Implen Inc., Germany). Three independent experiments were performed in triplicate.

**Statistical analysis**. All statistical analyses were performed using SPSS (v16). In box plots, data points more than 1.5X the interquartile range below or above the first or third quartile, respectively, are plotted as outliers. Real-time PCR data and integrated densities from western blotting and Confocal microscopy and CFU from agar plates were tested for normal distribution, and differences were compared using one-way analysis of variance (ANOVA) with Bonferroni's correction and 95% confidence intervals. $P$-values of <0.05 were considered significant.

## Data availability

The structures have been deposited in the Protein Data Bank under accession numbers 6BQS [https://doi.org/10.2210/pdb6bqs/pdb] and 6CRL [https://doi.org/10.2210/pdb6crl/pdb]. Assigned chemical shifts have been deposited in the Biological Magnetic Resonance Data Bank under accession number 27313 [https://doi.org/10.13018/BMR27313]. Other data are available upon reasonable request from the corresponding authors.

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

## Acknowledgements

The authors thank B. C. M. Yap and S. Dingsdag (Institute of Dental Research, Westmead Centre for Oral Health) for providing P8. SAXS experiments were performed using the SAXS/WAXS beamline at the Australian Synchrotron, ANSTO, with the assistance of Drs Nigel Kirby and Tim Ryan. HusA, HusA-Y164A and haemin samples for Raman spectroscopy were prepared by Florine Chretien and Raman experiments were performed at Sydney Analytical Core Research Facility with the assistance of Drs Anna-Maria Welsch and Joonsup Lee. We thank Dr Ryan Chai and Dr William Bubb for helpful comments on the manuscript. This project was supported by the Westmead Centre for Oral Health, Western Sydney Local Health District, Sydney Australia to N.H., The Schwartz Foundation to J.G., and a Faculty of Science, University of Sydney Seed Funding Grant 2016 to A.K.

## Author contributions

J.-L.G. and N.H. conceived the study; N.H. provided overall guidance to the study. J.-L.G., A.H.K. and D.A.G. led the various experiments and prepared the manuscript. J.-L.G., X.Z., K.-A.N., P.Y. and D.H. performed and analysed results for the biological and cellular assays; J.-L.G., A.Y. and A.H.K. performed and analysed results for the binding studies; B.M.H., D.A.T.C., D.A.G. and A.H.K. performed and analysed results from the NMR experiments and J.H. assisted with NMR spectra collection; D.A.G. determined the NMR structure and performed the docking studies; J.T. and A.H.K. performed and analysed results for the SAXS studies; A.H.K., J.-L.G. and D.A.G. performed and analysed results for the Raman studies. All authors except for P.Y., B.M.H., D.A.T.C. and J.H. proofread and suggested changes to the manuscript. All authors approved the manuscript.

## Additional information

**Competing interests:** The authors declare no competing interests.

