## [Peer Review File · Nature Communications]

Reviewers' comments:

Reviewer #1 (Remarks to the Author):

Manuscript # NCOMMS-17-30538-T

„Novel structural properties of a haemophore-like protein offer an avenue for targeted elimination of the pathogen *Porphyromonas gingivalis*“

Jin-Long Gao et al.

This manuscript further describes HusA protein. Although very sophisticated methods were used and interesting results were gained, this study requires more work to be convincing. The conclusions drawn (mainly regarding the possibility that porphyrin-metronidazole conjugate could be used to treat *P. gingivalis* infection) are very strong but not supported enough by the data presented.

Specific comments:

- How is this possible that *P. gingivalis* grows in the culture medium lacking iron (dipyridyl added) and supplemented with porphyrins without metal inside the porphyrin ring (Fig 1, panel A) ? What serves as an iron source allowing *P. gingivalis* to grow ? Were the bacteria starved before these experiments ? Please explain this more carefully.
- In this manuscript, there is no relation to the previous manuscript describing HusA properties, especially to conclusions that HusA binds dimeric heme. Results obtained from the former and the present manuscript are quite different. Please discuss this in more detail.
- Is there any possibility that HusA can bind heme but preferentially under reducing conditions ? Did the authors try to perform NMR analysis under reducing conditions ? There is a lot of data in the manuscript but they are provided in a way difficult to follow. It would be better to present the data as a regular research article not a communication.
- Bramanti and Holt (1991) and Olczak et al. (2012) suggested that porphyrins present in the culture medium together with iron could be used for heme formation by unknown proteins with a chelatase activity. Could it be possible that HusA might play such a function ? Please discuss this.
- ¹H NMR analysis should be described in more detail (some data are shown in Supplementary Figure 10 and 12 but are not properly described in the legend to these figures or in the main body of the manuscript). Could you assign particular peaks to porphyrin regions bound to HusA ? Is this possible to conclude (based on the NMR spectra obtained for the wild-type HusA with heme bound) about heme binding, assuming that such studies, similar to those published by Wojaczynski et al. (2011) or Wojtowicz et al. (2009) were done ?
- Reviewer is not an expert in NMR used to analyze protein structure and therefore do not make detailed comments regarding this method and data on the protein structure gained from these studies.
- The primary antibody used to detect *P. gingivalis* cells recognizes both cell attached and released proteins (DeCarlo and Harber, 1991). Is this possible to differentiate between these two fractions in vivo in the epithelial cell (Figure 1C) ? What we can see – *P. gingivalis* cells or both the protein associated with the bacterial cell and released into the epithelial cell environment ? Similar process was shown by Olczak et al. (2015) for HmuY.
- Please use “16S rRNA” throughout the manuscript (not 16s, 16S).
- Please add more detailed legend to the Supplementary Fig 17.
- Please change description “haem-binding site” by “porphyrin-binding site” according to the conclusion (page 4, lines 26-27) “Together, these results suggest that the porphyrin ring but not metal coordination is the major determinant of binding specificity.”
- Please add reference to the method of mutant construction (page 2, lines 24-25).
- Fig. 1 and Supplementary Fig. 2 – there is no difference in MW between HusA associated with the cell and the soluble protein.
- Spelling corrections are required:
 - a) page 3, line 22 (tetratricopeptide),
 - b) page 9, line 15 (Stratagene),
 - c) page 18, line 10 (SpectraMax),

d) legend to Fig. 1 (page 21, line 7) – different porphyrins.

Reviewer #2 (Remarks to the Author):

The paper reports a novel approach to combat periodontitis caused by *Porphyromonas gingivalis*. The HusA protein, that is essential for bacterial survival, has been used to deliver a porphyrin tethered antibiotic that results in intracellular elimination of the bacteria. The solution structure of the protein has been determined and a low affinity hydrophobic binding site for metallated and non-metallated porphyrins has been mapped.

The paper is original and of interest, however I believe there are some details that need clarifying in relation to analysis of the NMR data and the P8-HusA interaction before publication.

The NMR structure determination appears of good quality and the NMR family appears consistent with the relaxation data. However, I noticed that the NOESY data was recorded at 308K and the relaxation data at 298K. I understand assignment was aided by recording spectra at these two different temperatures, but I would like to know why the structure and relaxation data were not recorded at the same temperature. Also 5s for the recycle delay for the hetNOE experiments is only 5xT1, 10xT1 is normally required for these experiments. I would also like to see a PDB code included.

The chemical shift mapping was carried out at 298K and a number of resonances were reported as being lost. Looking at the spectra and the supplementary data I could not easily determine which resonances were lost (the BMRB data is being held until publication) and which resonances were unaffected. Although Fig 2C has this data I could not easily extract this information. As Hemin and Zn-PPIX affected similar residues the authors suggest that the paramagnetic effect is minimal and exchange broadening is giving rise to resonance loss or peak intensity reduction. To confirm this the authors should record HSQC spectra at lower temperatures to see if the peaks reappear shifted. What happened at 308K? What happened to the peaks when the NNNA mutant was titrated? Also, the DPIX HSQC spectrum (Fig S13) was recorded at a 3 molar excess of ligand and here it appears that many peaks are shifted and there is minimal resonance loss. What did the Hemin and Zn-PPIX spectra look like at a 3 molar excess of ligand?

Did the authors need to assign the porphyrin "bound" HusA HSQC spectra or did they pick the closest peaks to the apo-form? These questions are also related to the statement that DPIX binds in a similar manner to hemin and ZnPPIX, this may be in fact true, but just by looking at the spectra I cannot directly conclude this as the ligand concentration is very different for DPIX.

The docking experiments require accurate protonation states, I don't know how accurate the open babel program is, maybe the authors could confirm the states using propka/apbs.

For the P8 binding experiments I would like Fig3C to include hemin or PPIX for comparison. Also, was chemical shift mapping carried out for P8 with HusA? This would confirm binding and reinforce the conclusion that HusA is binding and transporting the tethered antibiotic.

Ending on a very minor note - abovementioned or above-mentioned?

Reviewer #3 (Remarks to the Author):

The authors describe the purification and characterization of a novel fold hemophore from *P. gingivalis*. Much of the data in Fig. 1 is confirmation of previous data published by the authors in JBC in 2010. The novel new data is the solution structure which shows a distinct fold from that of

previously described hemophores from *P. gingivalis* or the bacterial hemophores of *Serratia* or *Pseudomonas*. This is novel but the authors do not discuss this in any great detail or how this relates to specificity of binding to the receptor. Interestingly the purified recombinant hemophore does not contain heme as has been described for other hemophores despite similar K_D values. The fluorescence quenching experiments are not described in any detail and there may be some concern with the emission spectra for Trp between 300 and 500 nm given the overlap with the excitation of the porphyrins? Was this accounted for?

Consistent with a hydrophobic binding pocket mutations that affect binding are largely hydrophobic with the exception of Y164. The authors again do not address a potential role of Tyr-164 as a ligand in great detail or completely rule out that there is coordination. The spectroscopy performed is relatively weak no reports on stoichiometry or shift in the Soret peaks on mutation or what this means in terms of coordination etc. It is not clear to me that the authors have definitively shown that Y164, which seems poised to coordinate to the heme, does not do so. A moderate loss in binding affinity on removal of the ligand is not evidence of no coordination. Furthermore, hemophores bind PPIX relatively tightly given the fact that hydrophobic interactions with the porphyrin macrocycle drive the binding with the coordination contributing relatively little to the free energy of binding. Indeed, in the HasA coordination mutants a similar decrease in affinity is observed as the major contribution to binding are the hydrophobic interactions (see the work of Rivera et al). A more detailed spectroscopic analysis of the heme coordination chemistry would strengthen the manuscript.

A positive aspect of the manuscript is the fact the authors show convincingly that *husA* plays a role in the pathogenicity of *P. gingivalis*. They further use a Trojan horse approach which has been employed in siderophore uptake systems appears to show efficacy in inhibiting *P. gingivalis*. However, studies were not performed with a combination of PPIX and metronidazole as the control rather than metronidazole alone. Is it possible that PPIX or DPIX sensitizes the cells (especially in iron restricted conditions)? These controls are important to perform.

Overall the manuscript provides new insight into the structure of HusA and presents preliminary data on heme or porphyrin binding. However, the authors do not place this in perspective of the field nor do they rule out metal coordination based on the current studies. Similarly, the Trojan horse approach of delivering metronidazole shows some promise, however, some controls are lacking especially given the porphyrins can sensitize the cells to treatment.

Reviewer #1 (Remarks to the Author):

This manuscript further describes HusA protein. Although very sophisticated methods were used and interesting results were gained, this study requires more work to be convincing. The conclusions drawn (mainly regarding the possibility that porphyrin-metronidazole conjugate could be used to treat *P. gingivalis* infection) are very strong but not supported enough by the data presented.

Specific comments:

- How is this possible that *P. gingivalis* grows in the culture medium lacking iron (dipyridyl added) and supplemented with porphyrins without metal inside the porphyrin ring (Fig 1, panel A)? What serves as an iron source allowing *P. gingivalis* to grow? Were the bacteria starved before these experiments? Please explain this more carefully.

Response: We agree with the reviewer that *P. gingivalis* requires both iron and porphyrin to survive. Iron is naturally present in the complex medium, enriched tryptic soy broth (eTSB), used in this study. The iron concentration of TSB medium has been previously determined at 0.74 µg/mL by atomic absorption spectrophotometry (de Oliverira Moreira L., et al., 2003 AEM) which requires approximately 500 µM dipyridyl to chelate all available iron in the medium (Scharfman AH 1996 I&I). The dipyridyl used in this study is 100 µM and the remaining iron provided by the TSB medium is sufficient to support *P. gingivalis* growth. The bacteria were starved by passaging in haemin free eTSB medium without dipyridyl as described in the Methods (Page 12, lines 27-29). The starvation was applied to deplete the bacteria's stores of haemin and porphyrin, in particular the accumulated black pigment, haemin, carried by the colony from blood agar. As a porphyrin auxotroph, *P. gingivalis* cannot grow in the last passage mainly due to the depletion of porphyrin in the medium, even though iron is bioavailable in the eTSB medium. To further prove this, we conducted the growth experiment for *P. gingivalis* W83 with titration of dipyridyl as shown in Supplementary Fig 2. The recovery of growth of *P. gingivalis* W83 after starvation was observed in the presence of up to 200 µM dipyridyl while *P. gingivalis* failed to recover in the presence of 500 µM and above.

- In this manuscript, there is no relation to the previous manuscript describing HusA properties, especially to conclusions that HusA binds dimeric heme. Results obtained from the former and the present manuscript are quite different. Please discuss this in more detail.

Response: We agree with the reviewer's comments and have included more in depth discussion on this topic. We have now added in the discussion "While the data from *in vitro* haemin titration experiments fit well to a 1:1 binding model, the hydrophobic binding groove could potentially also allow HusA to accommodate dimeric haem from *P. gingivalis* pigmentation extract (Gao, et al., 2010, JBC)." Also "While we have previously reported the HusA:haem binding to be of high affinity ($K_d \sim 2$ nM), our current investigations suggest that the actual binding affinity is far lower and in the low micromolar range. The micromolar binding affinity is more consistent with the intermediate exchange observed in the NMR titration experiment, and the inability to consistently purify the HusA:haemin complex using size-exclusion chromatography." In the previous paper, the concentrations used were much lower leading to larger errors and a plateau was not adequately achieved leading to inaccuracies in estimation of the binding affinity.

- Is there any possibility that HusA can bind heme but preferentially under reducing conditions? Did the authors try to perform NMR analysis under reducing conditions?

Response: NMR and SAXS experiments were performed under reducing (TCEP) and non-reducing buffer conditions, giving essentially the same results (e.g. see SAXS table for comparison between HusA:haem under reduced and oxidised conditions), suggesting that cysteine oxidation does not occur readily, or does not substantially affect function. To investigate the role of iron reduction/oxidation state, which is the reviewer's main concern, we performed UV-vis binding experiments with Fe(II) haem reduced with sodium dithionate. Spectral changes occur after mixing HusA with Fe(II) haem, suggesting that binding is occurring. This experiment does not address the relative affinity of HusA for Fe(III) vs Fe(II) haem, but does demonstrate that either oxidation state is compatible with complex formation. Analysis of the haemin binding site reveals that Y164 is the only strong candidate for an axial haem ligand. Notably, haem(s) with a tyrosine ligand typically have a large negative midpoint reduction potential and so have a strong preference for binding Fe(III), rather than Fe(II), haem(s). We have updated the manuscript with additional experiments (Figure 2G, H) that support the conclusion that HusA binds haem and non-metallated porphyrins with little or no contribution from a protein ligand to the haem iron (also discussed further below). In summary, we think that HusA will bind equally well to Fe(II)/(III)-PIX.

There is a lot of data in the manuscript but they are provided in a way difficult to follow. It would be better to present the data as a regular research article not a communication.

Response: We agree with the reviewer's comments and have now re-written the manuscript as a full article instead of a letter, and have endeavoured to supply more explanation and improve the logical flow.

- Bramanti and Holt (1991) and Olczak et al. (2012) suggested that porphyrins present in the culture medium together with iron could be used for heme formation by unknown proteins with a chelatase activity. Could it be possible that HusA might play such a function? Please discuss this.

Response: The reviewer raises an interesting possibility, that HusA contributes to a ferrochelatase activity of *P. gingivalis* that has been reported in the literature. The reverse ferrochelatase activity of recombinant HusA and *P. gingivalis* outer membrane extract was examined and the results are presented in new supplementary Fig 17. The methods are detailed in the Supplementary methods section. As expected, the outer membrane extract, but not the rHusA, demonstrated reverse ferrochelatase activity.

- ¹H NMR analysis should be described in more detail (some data are shown in Supplementary Figure 10 and 12 but are not properly described in the legend to these figures or in the main body of the manuscript). Could you assign particular peaks to porphyrin regions bound to HusA? Is this possible to conclude (based on the NMR spectra obtained for the wild-type HusA with heme bound) about heme binding,

assuming that such studies, similar to those published by Wojaczynski et al. (2011) or Wojtowicz et al. (2009) were done ?

Response: In the main text we have added “No haem hyperfine shifted signals (i.e. below –2 or above 10 ppm where protein signals are found) can be seen ¹H 1D NMR spectra recorded at temperatures from 298 K to 328 K (Supplementary Figure 15), providing no evidence for iron coordination by HusA side chains. This is in contrast to the HmuY haemophore from *P. gingivalis* for which hyperfine shift signals could be observed upon the ligation of haem, iron(III) mesoporphyrin IX and iron(III) deuteroporphyrin IX by HmuY (Wojaczynski et al. (2011) Wojtowicz et al. (2009)).”

- The primary antibody used to detect *P. gingivalis* cells recognizes both cell attached and released proteins (DeCarlo and Harber, 1991). Is this possible to differentiate between these two fractions in vivo in the epithelial cell (Figure 1C) ? What we can see – *P. gingivalis* cells or both the protein associated with the bacterial cell and released into the epithelial cell environment ? Similar process was shown by Olczak et al. (2015) for HmuY.

Response: We thank the reviewer for seeking clarification on this point. The primary antibody used in this study is commonly used to probe against gingipain proteins, which are highly expressed on the outer membrane surface and vesicles produced by *P. gingivalis*. The average size of released vesicles is approximately 50 nm (Xie H, Future Microbiol 2015; Olsen I and Progulske-Fox A, J Oral Microbiol, 2015) where the released proteins would be expected to be even smaller in size. The average diameter of *P. gingivalis* cells is 1-2 μm, which is about 20 times larger than the vesicles. Hence the high magnification image shown in Fig 1Ciii represented invading *P. gingivalis* cells rather than released proteins. Similarly, the anti-HmuY antibody also targeted the HmuY haemophore, which exists as both bacterial cell associated form and released form. We agree that anti-HmuY antibody could be another option.

- Please use “16S rRNA” throughout the manuscript (not 16s, 16S).

Response: We have changed this throughout (see Pages 14, 24)

- Please add more detailed legend to the Supplementary Fig 17.

Response: We thank the reviewer for comments. A detailed description of the Supplementary Fig 24 (previously as Supplementary Fig 17) has been provided in the revised manuscript.

- Please change description “haem-binding site” by “porphyrin-binding site” according to the conclusion (page 4, lines 26-27) “Together, these results suggest that the porphyrin ring but not metal coordination is the major determinant of binding specificity.”

Response: We thank the reviewer’s comment and have substituted “haem/porphyrin-binding site” in the relevant sections.

- Please add reference to the method of mutant construction (page 2, lines 24-25).

Response: We have added text (page 12) “Expression plasmids of various site-directed mutagenic recombinant proteins were constructed using a modified version of the site-directed, ligase-independent mutagenesis method described previously by Chiu et al (*J Microbiol Methods* **73**, 195-8 (2008)).”

- Fig. 1 and Supplementary Fig. 2 – there is no difference in MW between HusA associated with the cell and the soluble protein.

Response: HusA was detected in the *P. gingivalis* outer membrane extract, extracellular vesicles, and extracellular vesicle-free medium. HusA was predicted to possess a lipoprotein signal peptide by LipoP 1.0 (<http://www.cbs.dtu.dk/services/LipoP/>). HusA may be able to anchor in the bacterial outer membrane via a lipid modified residue, however it is quite likely that this would not substantially alter the protein mobility on SDS-PAGE. It has precedent that *P. gingivalis* HmuY haemophore was also reported to

present as a membrane associated form and soluble form with identical molecular weight (as presented in Fig4B, Olczak T, et al, BMC Microbiology, 2000).

- Spelling corrections are required:

- a) page 3, line 22 (tetratricopeptide),
- b) page 9, line 15 (Stratagene),
- c) page 18, line 10 (SpectraMax),
- d) legend to Fig. 1 (page 21, line 7) – different porphyrins.

Response: We thank the reviewer for the time, effort and care taken to provide these corrections. The spelling errors have been corrected.

Reviewer #2 (Remarks to the Author):

The paper reports a novel approach to combat periodontitis caused by *Porphyromonas gingivalis*. The HusA protein, that is essential for bacterial survival, has been used to deliver a porphyrin tethered antibiotic that results in intracellular elimination of the bacteria. The solution structure of the protein has been determined and a low affinity hydrophobic binding site for metallated and non-metalled porphyrins has been mapped.

The paper is original and of interest, however I believe there are some details that need clarifying in relation to analysis of the NMR data and the P8-HusA interaction before publication.

The NMR structure determination appears of good quality and the NMR family appears consistent with the relaxation data. However, I noticed that the NOESY data was recorded at 308K and the relaxation data at 298K. I understand assignment was aided by recording spectra at these two different temperatures, but I would like to know why the structure and relaxation data were not recorded at the same temperature. Also 5s for the recycle delay for the hetNOE experiments is only 5xT1, 10xT1 is normally required for these experiments.

Response: We have now added to the results “NMR data were collected at two temperatures (298 K and 308 K) to assist with resonance assignment, for example by resolving signal overlaps. The final NOESY spectra used for NMR structure calculations were collected at 308 K, which gave the best compromise between signal intensity and spectral dispersion/completeness (temperatures in range 278–323 K were investigated).” In addition, ¹⁵N relaxation data (T₁, T₂, heteronuclear NOE) have been acquired at both 298 K and 308K, and the results are shown in Supplementary Figure 7. There are no substantial differences between the two data sets, indicating that the shift 298 K to 308 K does not significantly alter internal dynamics.

Indeed for quantitate measurements of relaxation time constants, 10xT1 is normally required. However, for comparing relative relaxation properties of different residues in a protein (as are done in this study), 5xT1 is sufficient and routine. Please see <http://pubs.acs.org/doi/pdfplus/10.1021/bi00138a005> and [http://www.nmr2.buffalo.edu/nesg.wiki/Measuring_15N_T1_and_T2_relaxation_times_\(Bruker\)](http://www.nmr2.buffalo.edu/nesg.wiki/Measuring_15N_T1_and_T2_relaxation_times_(Bruker))

I would also like to see a PDB code included.

Response: PDB codes, 6BQS and 6CRL, have been included in Supplementary Table 1, NMR structure statistics

The chemical shift mapping was carried out at 298K and a number of resonances were reported as being lost. Looking at the spectra and the supplementary data I could not easily determine which resonances were lost (the BMRB data is being held until publication) and which resonances were unaffected. Although Fig 2C has this data I could not easily extract this information. As Hemin and Zn-PPIX affected

similar residues the authors suggest that the paramagnetic effect is minimal and exchange broadening is giving rise to resonance loss or peak intensity reduction. To confirm this the authors should record HSQC spectra at lower temperatures to see if the peaks reappear shifted. What happened at 308K? What happened to the peaks when the NNNA mutant was titrated? Also, the DPIIX HSQC spectrum (Fig S13) was recorded at a 3 molar excess of ligand and here it appears that many peaks are shifted and there is minimal resonance loss. What did the Hemin and Zn-PPIX spectra look like at a 3 molar excess of ligand?

Response: We have added substantially more details that describe the NMR titration experiments. In general spectral quality for apo HusA is far poorer (with severe line broadening) at the lower temperatures so all experiments were carried out at 298 and/or 308 K. We have carried out hemin, DPIIX and P8 titrations at 298 K and 308 K with identical conclusions.

Overall, there are a common set of resonances that disappear for all porphyrin ligands we have tested, including the DPIIX-metronidazole conjugate (P8); however, as noted by the reviewer, there are also some resonances that shift in approximately fast exchange, only for some of the porphyrins (notably DPIIX and P8). An example of both behaviours is shown in Supplemental Figure 21 (P8 titration). Supplemental Figure 18 A now shows the HSQC of HusA:haemin and HusA:DPIIX at ~ 1:1 molar ratio and 303 K; the peak annotations show that a very similar set of resonances in both complexes have become weak or disappear (e.g., Q108, R110, T30, R102, V105, A60, A67), in addition, peaks are slightly shifted in the HusA:DPIIX spectrum relative to the HusA:haemin and HusA alone spectra. Supplemental Figure 18 B–D show graphs of chemical shift changes (CSPs) as well as intensity loss for haem analogues where CSPs are evident. We have added text “NMR titration analysis indicates that P8 binds at a similar site to haemin and DPIIX (Supplementary Figure 17B, C). Interestingly, titrations of both DPIIX and P8 into apo HusA resulted in chemical shift perturbations (CSPs) that approximated a fast exchange regime for a subset of residues more distant from the binding groove (Supplementary Figure 21), in addition to signal loss at early titration points for other residues. “

Did the authors need to assign the porphyrin “bound” HusA HSQC spectra or did they pick the closest peaks to the apo-form? These questions are also related to the statement that DPIIX binds in a similar manner to hemin and ZnPPIX, this may be in fact true, but just by looking at the spectra I cannot directly conclude this as the ligand concentration is very different for DPIIX.

Response: We monitored and followed peak positional changes (for peaks in fast exchange, e.g., as shown in Supplemental Fig 21) and disappearances (for peaks in intermediate and slow exchange) as the porphyrins were titrated in apo-HusA. Overall the number of peaks in the spectra decreased as porphyrin was added, suggesting that most peaks did not reappear even with 2–3 molar excess of ligand. Thus for the few peaks that did appear during titration, it would not be possible to obtain assignments using conventional sequential assignment methods.

The docking experiments require accurate protonation states, I don't know how accurate the open babel program is, maybe the authors could confirm the states using propka/apbs.

Response: Protonation states calculated with open babel or propka are the same. Note that preparation of the input files for Autodock Vina requires correct protonation states, but the docking algorithm uses an extensively tested empirical scoring algorithm, rather than a force field approach for which electrostatic potentials need to be accurately calculated.

For the P8 binding experiments I would like Fig3C to include hemin or PPIX for comparison. Also, was chemical shift mapping carried out for P8 with HusA? This would confirm binding and reinforce the conclusion that HusA is binding and transporting the tethered antibiotic.

Response: The haemin or PPIX have been included in the P8 binding experiment for comparison (Fig 4C). We have additionally carried out chemical shift mapping carried out for P8 with HusA

(Supplementary Fig 17C and Supplementary Figure 21). The results identify a similar binding site for P8, DPIIX and haemin

Ending on a very minor note - abovementioned or above-mentioned?

Response: We think both versions are used and are acceptable and will leave this to be decided upon during the typesetting process.

Reviewer #3 (Remarks to the Author):

The authors describe the purification and characterization of a novel fold hemophore from *P. gingivalis*. Much of the data in Fig. 1 is confirmation of previous data published by the authors in JBC in 2010. The novel new data is the solution structure which shows a distinct fold from that of previously described hemophores from *P. gingivalis* or the bacterial hemophores of *Serratia* or *Pseudomonas*. This is novel but the authors do not discuss this in any great detail or how this relates to specificity of binding to the receptor. Interestingly the purified recombinant hemophore does not contain heme as has been described for other hemophores despite similar KD values.

Response: This is an important point raised by the referee. HusA is related to the TPR fold family, which appears to be a robust scaffold to acquire novel binding functions, and has been co-opted to several unrelated binding functions in different pathogens (see for example review by Cerveny 2013 I&I). HusA is the only TPR-like protein that binds porphyrins and the implication is that it has acquired this function during evolution of the *Porphyromonas* species. The apparent selectivity for metal-free over metallated porphyrins suggests that HusA might function as part of the pathway for acquisition of free porphyrins derived from reverse chelatase activity on the extracellular pigment. The relatively low affinity of HusA is similar to affinity of another mammalian porphyrin binding protein, p22HBP and is consistent with HusA targeting porphyrin as a preferred ligand over haem. We have expanded the discussion in this regard.

The fluorescence quenching experiments are not described in any detail and there may be some concern with the emission spectra for Trp between 300 and 500 nm given the overlap with the excitation of the porphyrins? Was this accounted for?

Response: We have moved the method section that describes the fluorescence quenching experiments into Supplementary methods so more details about how the inner filter effect was corrected are included together with additional reference.

Consistent with a hydrophobic binding pocket mutations that affect binding are largely hydrophobic with the exception of Y164. The authors again do not address a potential role of Tyr-164 as a ligand in great detail or completely rule out that there is coordination. The spectroscopy performed is relatively weak no reports on stoichiometry or shift in the Soret peaks on mutation or what this means in terms of coordination etc. It is not clear to me that the authors have definitively shown that Y164, which seems poised to coordinate to the heme, does not do so. A moderate loss in binding affinity on removal of the ligand is not evidence of no coordination. Furthermore, hemophores bind PPIX relatively tightly given the fact that hydrophobic interactions with the porphyrin macrocycle drive the binding with the coordination contributing relatively little to the free energy of binding. Indeed, in the HasA coordination mutants a similar decrease in affinity is observed as the major contribution to binding are the hydrophobic interactions (see the work of Rivera et al). A more detailed spectroscopic analysis of the heme coordination chemistry would strengthen the manuscript.

Response: We thank the reviewer for encouraging a more thorough investigation into the coordination chemistry. As noted by the reviewer, the position of Tyr164 clearly warrants attention. Our initial NMR investigation failed to find any hyperfine shifts that would suggest protein side chain potentially ligating the haem iron (Supplementary Figure 15). We have performed additional experiments in the manuscript

that show UV-visible spectra of HusA:haem and mutants Y164A or Y164F are very similar, all with a typical signature of ferric spin equilibrium including an absorption band ≥ 600 nm that could arise either from a Tyr hydroxyl or a water ligand (Fig 2G). In each of HusA, Y164A and Y164F, the spectrum undergoes acid-alkaline transition with pKa $\sim 6-6.5$ (Fig 2H), and on the basis of this we assign the absorption band at >603 nm to be an axial water/hydroxide, and not an axial O ligand from the phenol group of Y164, which would show no titration in this pH range. As a demonstration of this we provide spectra and pH titration of two bacterial proteins, IsdH and IsdB, that ligand Fe(III) haem through Tyr side chains.

Regarding the loss (or otherwise) of binding affinity when a haem ligand is mutated, whilst this is not definitive evidence, the similar binding affinities displayed by HusA for metallated or non-metallated porphyrins (in fact the latter actually bind more strongly), and the small changes (2 \times) in binding affinity of the Y164A mutant, are at least consistent with the spectroscopy, in suggesting that iron coordination cannot contribute substantially to ligand binding. Indeed, for the HasA system mentioned by the reviewer, mutation of the primary Tyr haem ligand does cause a substantial (400 \times) drop in haem affinity (Deniau et al. Biochemistry, 2003), although mutation of a second axial His causes only moderate 2–5 fold drop.

A positive aspect of the manuscript is the fact the authors show convincingly that husA plays a role in the pathogenicity of *P. gingivalis*. They further use a Trojan horse approach which has been employed in siderophore uptake systems appears to show efficacy in inhibiting *P. gingivalis*. However, studies were not performed with a combination of PPIX and metronidazole as the control rather than metronidazole alone. Is it possible that PPIX or DPIX sensitizes the cells (especially in iron restricted conditions)? These controls are important to perform.

Response: We thank the reviewer for this critical comment. The additional control has been performed and the results are presented in the updated in Figure 4D (metronidazole + PPIX and metronidazole + DPIX) as suggested by the reviewer. We also performed growth analysis to examine the possibility that PPIX or DPIX sensitizes *P. gingivalis* cells under iron replete and iron limited conditions. As shown the Supplementary Fig 23, we did not observe any significant synergistic effects between porphyrins and metronidazole under the experimental conditions.

Overall the manuscript provides new insight into the structure of HusA and presents preliminary data on heme or porphyrin binding. However, the authors do not place this in perspective of the field nor do they rule out metal coordination based on the current studies. Similarly, the Trojan horse approach of delivering metronidazole shows some promise, however, some controls are lacking especially given the porphyrins can sensitize the cells to treatment.

Response: We thank the reviewer for this comment. We have made substantial additions to the manuscript including comparisons to other haem and porphyrin binding proteins. Multiple lines of evidence are presented that are consistent with absence of meaningful metal coordination. The important controls against cellular sensitization by porphyrin and concerted action of porphyrin and non-conjugated antibiotic have been performed.

Reviewers' comments:

Reviewer #1 (Remarks to the Author):

The Authors responded to all the Reviewer's comments and significantly improved the manuscript content.

Reviewer #2 (Remarks to the Author):

The authors have managed to answer all the queries and points of discussion that I raised after reading the first draft. Existing data has been clarified and new data included to improve the manuscript. I believe the paper is now suitable for publication.

Reviewer #3 (Remarks to the Author):

The revised manuscript is improved, but remains inconclusive and incomplete with regard to key findings. Specifically, significant issues remain about interpretation and validation of key results. The manuscript still comes across as being poor where P8-HusA interaction is concerned as the majority of the data revolves around HusA-hemin or HusA-DPIX interactions. The potential impact of the findings can be improved by focusing on the P8 binding to HusA.

Remaining Concerns:

1) Fig. 1A still needs confirmation with the inclusion of two additional panels.

(a) Effect of 0.5 μ M P8 added to the culture prior to the commencement of growth.

(b) Effect of adding 0.5 μ M P8 to actively growing cultures of *P. gingivalis* W83

2) Fig. 1C: What happens when P8 (0.5 μ M) is added? Do the cells lyse? Why is the green fluorescence not uniform in panel 'i'?

4) HSQC spectra for the interaction of HusA with P8 should be provided (analogous to Supplementary Fig. 18A, where the authors have done a nice job of showing the data down to noise level). Specifically, HSQC superpositions of DPIX and P8 would make for direct comparison. Similarly, the authors would also need to show an HSQC comparison of HusA alone and bound to P8. Indeed, Supplementary Fig. 21 is helpful in analyzing the exchange regimes of a couple of residues, but it doesn't give a global picture of the changes. Similarly, Supplementary Fig. 18C only represents the authors' interpretation and not the raw data, which should be available for independent reproduction in the future.

5) On p. 42 of the Supplementary Material, the authors describe their methodology for determining binding constants via fluorescence quenching assays. Supplementary Tables S3 and S4 reveal the K_d values for HusA (and mutants) - hemin and HusA-porphyrin analogs. Presumably these data were obtained via non-linear least squares fit to the described quadratic. However, both Figs. 3E and 4C shows raw data and NOT the results from the fitting of the data. This is important to show given that neither hemin nor P8 reach saturation and the DPIX data does not go beyond 10 μ M. Furthermore, Fig. 2F appears biphasic and suffers from the same issues pointed out above for Figs. 3E and 4C. These discrepancies may explain why the authors originally reported a nanomolar binding affinity for HusA-hemin interaction when in reality the K_d values are orders of magnitude off! The authors need to clarify this in more detail because if the fits are not correct, the authors are likely to introduce uncertainties into the primary literature.

6) There is a discrepancy between Fig. 2D and 4B. The former only shows "residues experiencing a large reduction of NMR signal" for hemin or Zn(II)PPIX binding, whereas the latter includes

chemical shift changes for the P8 interaction. It is unclear whether these chemical shift perturbations are also observed when HusA is associating with hemin or Zn(II)PPIX. If there are chemical shift differences between hemin vs P8, it is necessary to specifically document these, as it may suggest conformational changes or additional porphyrin binding sites. Also, it is hard to be sure what is going on and if there are differences between even Zn-PPIX and hemin based on Supplementary Fig. 10.

7) Binding stoichiometry: The authors have used a fixed [HusA] (1 μ M) while varying hemin, DPIX, or P8 (Fig. 4C). If the curve fits were done correctly, they should not only provide K_d values, but also the ligand stoichiometry. To a first approximation, if one were to assume that DPIX interacts with a single site on HusA, then by extension it appears that there may be more sites for porphyrin binding when hemin or P8 are titrated with HusA. The manuscript is confusing in its use of the terms, such as "sub-stoichiometric" (cf. p. 6, line 16). What exactly are the authors trying to achieve through the use of variable "molar equivalents" of porphyrin molecules?

Finally, concerns still remains with regard to hemin binding. The authors do numerous experiments including mutagenesis and comparisons to other proteins such IsdH but these are all indirect and fraught with difficulty by changing the heme environment. The authors still have not convincingly performed any spectroscopy that could give a more detailed fingerprint such as resonance Raman or magnetic circular dichroism.

Similarly, the binding of uroporphyrin and coproporphyrin does not show differences in spectrum. How was this shown to be specific? Furthermore, this work if all of the issues are addressed can stand on its own without resorting to discussions pertaining to the controversial "reverse ferrochelatase" activity, which I am unclear as to why the authors needed to include.

Rigor, reproducibility, transparency, and presentation issues:

1) Why does the scaled fluorescence not approach zero in Fig. 4C post inner-filter effect corrections? One possibility is that the corrections have not been properly accounted for. This is evident in the green curve shown in Fig. 3E. Have the authors accounted for Zn(II)PPIX having different fluorescence characteristics than the other porphyrins?

2) Re: Figs. 3E and 4C, it would be essential to provide not only the fits, but also a panel showing the goodness of fit, i.e. residual plot.

3) SEM should include error bars for data in Fig 4C, which currently would be interpreted as being derived from a $n=1$ data set

4) Supplementary Fig. 3A and B appear to have been sliced and combined from different panels. If this is the case then the entire unmodified gel/blot should be shown in the Supplement for the sake of transparency.

5) What do the authors mean by " ≥ 2 standard deviations" in the context of Figs. 2D & 4B? Precisely, how were these calculated? It is ok to use the terms, such as "concave" and "convex", but it'd be important to point out to the reader that the former can be generated from the latter through a 180 degree rotation.

6) Title of Fig. 2 is misleading. It is not just the porphyrin binding site, but (as shown) hemin binding pocket as well.

7) What do the blue and red spheres represent in Fig. 2D?

8) Data should be included in Fig. 4F to illustrate the consequences of 1 μ M P8, which W83 is susceptible to (p. 31, line 18). This data would provide the susceptibility index.

9) What is the basis for the differences in the metronidazole conformations in the two conformers shown (Fig. 4B)? Is this a consequence of computational modeling or based on experimental data? This should be stated.

10) Fig. 4F should be moved to the Supplementary Data. The authors should generate a similar figure as in 4B based on their chemical shift data, showing the interaction between P8 and HusA.

11) What is the significance of Supplementary Fig. 7 where dynamics data is provided exclusively for the apo-protein. How do the parameters change upon P8 addition?

Mechanistic questions:

a) As administered, metronidazole is a prodrug and requires reductive activation for achieving its cytotoxic effects. Reductive inactivation of the nitro function results in the imidazole group cleavage. Alternatively, metronidazole is capable of interfering with chemiosmotic coupling and energy conservation. It'd be important for the authors to at least speculate on whether covalently fused metronidazole is capable of functioning like its free counterpart or if it is released? The experiments with metronidazole plus PPIX or DPIX are not the same as with the P8 scaffold (including the Boc protecting group) plus metronidazole. As *P. gingivalis* is a heme auxotroph, would the authors expect P8 to impact other proteins?

c) *P. gingivalis* is a nanoaerobe, do the authors expect P8 to block oxygen respiration? Are all the growth, viability, and potentiation assays performed under strict anaerobiosis? It appears to be the case, but the authors may wish to clarify whether or not O₂ contamination is likely during the course of the experiments. Notably, does *P. gingivalis* take up heme during anaerobic conditions or is it a consequence of triggering aerobic respiration.

Reviewer #3 (Remarks to the Author): The revised manuscript is improved, but remains inconclusive and incomplete with regard to key findings. Specifically, significant issues remain about interpretation and validation of key results. The manuscript still comes across as being poor where P8-HusA interaction is concerned as the majority of the data revolves around HusA-hemin or HusA-DPIX interactions. The potential impact of the findings can be improved by focusing on the P8 binding to HusA.

Remaining Concerns:

1) Fig. 1A still needs confirmation with the inclusion of two additional panels.

(a) Effect of 0.5 μM P8 added to the culture prior to the commencement of growth. (b) Effect of adding 0.5 μM P8 to actively growing cultures of *P. gingivalis* W83

Response: The growth curve studies, including addition of 0.5 μM P8 prior to the commencement of growth as well as in the mid-log growth, have been conducted as suggested by Reviewer 3. In the presence of dipyriddy, 0.5 μM P8 retards the growth of wild type *P. gingivalis* W83 when added prior to the commencement of growth (low bacteria density). *P. gingivalis* W83 fails to growth in the presence of 2.5 μM P8 under the same condition. When added to the actively growing culture (high bacteria density) of *P. gingivalis* W83, both 0.5 μM and 2.5 μM P8 retard growth with the 2.5 μM P8 treatment achieving a relative lower density at stationary phase. The MIC of P8 is correlated to the initial bacteria densities in the inoculum. Udekwu et al. have reported that higher concentration of antibiotics is required to achieve the MIC when treating higher densities of bacteria albeit with variation according to class of drug (Udekwu KI et al, J Antimicrob Chemother. 2009; 63(4):745-757).

As P8 is introduced in the later section of the main text, we have presented the results as Supplementary Fig 22.

2) Fig. 1C: What happens when P8 (0.5 μM) is added? Do the cells lyse? Why is the green fluorescence not uniform in panel 'i'?

Response: P8 does not induce cell toxicity at concentrations above 0.5 μM , as illustrated in Supplementary Fig. 23D. As shown in the Confocal Microscopic images of Supplementary Fig. 23C, epithelial cells remain intact in the presence of 20 μM P8.

The primary antibody used in this study is commonly used to probe against gingipain proteins, which are highly expressed on the outer membrane surface and vesicles produced by *P. gingivalis*. The green fluorescence indicates the presence of both bacteria and bacterial vesicles. The average diameter of *P. gingivalis* cells is 1-2 μM , which is about 20 times larger than the vesicles. Hence the green fluorescence is not uniform in panel "i".

3) HSQC spectra for the interaction of HusA with P8 should be provided (analogous to Supplementary Fig. 18A, where the authors have done a nice job of showing the data down to noise level). Specifically, HSQC superpositions of DPIX and P8 would make for direct comparison. Similarly, the authors would also need to show an HSQC comparison of HusA alone and bound to P8. Indeed, Supplementary Fig. 21 is helpful in analyzing the exchange regimes of a couple of residues, but it doesn't give a global picture of

the changes. Similarly, Supplementary Fig. 18C only represents the authors' interpretation and not the raw data, which should be available for independent reproduction in the future.

Response: The requested superpositions of DPIX and P8, and with HusA alone are shown in Supplementary Figure S21A, in addition to superposition of haemin, DPIX and HusA in Supplementary Figure S18. The old Figs S18C and S21 have been replaced with raw HSQC data provided in Supplementary Figure 21B that now gives a global picture of the chemical exchange regimes for different sets of residues.

4) On p. 42 of the Supplementary Material, the authors describe their methodology for determining binding constants via fluorescence quenching assays. Supplementary Tables S3 and S4 reveal the K_d values for HusA (and mutants) - hemin and HusA-porphyrin analogs. Presumably these data were obtained via non-linear least squares fit to the described quadratic. However, both Figs. 3E and 4C shows raw data and NOT the results from the fitting of the data. This is important to show given that neither hemin nor P8 reach saturation and the DPIX data does not go beyond 10 μ M. Furthermore, Fig. 2F appears biphasic and suffers from the same issues pointed out above above for Figs. 3E and 4C. These discrepancies may explain why the authors originally reported a nanomolar binding affinity for HusA-hemin interaction when in reality the K_d values are orders of magnitude off! The authors need to clarify this in more detail because if the fits are not correct, the authors are likely to introduce uncertainties into the primary literature.

Response: The non-linear least squares fittings were carried out directly on fluorescence intensities after correction of inner filter effect as described in the Methods and carried out in Origin7 and Origin2016. The fitted curves and residuals have now been included as Supplementary Fig S14. The fluorescence data shown in the main text are indeed data points after correction for inner filter effect and scaling to one at the start of the assay – this is simply to allow an easy comparison/plotting of different experiments (e.g. HusA and mutants; haem and porphyrin analogues) on the same graph. The lines connecting the data points in Figure 2F have been removed to avoid their misinterpretation as fitted curves. We have repeated the DPIX assay to saturation and the new dataset is shown in the Fig. S14 and this does not significantly change the K_d (new value is 0.31 ± 0.06 , old value was $0.36 \pm 0.09 \mu$ M). However, we have left Fig. 2F and Table S3 as they are to allow a more direct comparison (e.g. the titrations in the set are done with the same batch of HusA protein or haem stocks).

In some cases, the inner filter effect/solubility of porphyrins precludes going to higher concentrations to determine the absolute binding affinities. The low μ M binding affinities that we have estimated are consistent with observations of a mixture of intermediate and fast exchange regimes in NMR titration experiments and the inability for the complexes to be purified on size-exclusion chromatography.

Our primary motivation for this set of experiment is not to determine the absolute binding affinities but to compare relative binding affinities with the aim of investigating the effect of mutations to probe binding residues and porphyrin specificity. Hence we prefer to present datasets that have used using the same haemin stock (for HusA and mutant comparisons) or batch of protein (for porphyrin analogue comparisons) since errors in concentration estimation of the stock can easily introduce errors in absolute binding affinity but the relative binding affinity still holds. The latter is important for our interpretation.

The P8 binding curve is indeed relatively flat and hence we have not fitted this data with the 1:1 binding model. We do not report the binding affinity of HusA to P8 but simply give an estimate of ~1–10 μM by comparison to other porphyrins.

5) There is a discrepancy between Fig. 2D and 4B. The former only shows "residues experiencing a large reduction of NMR signal" for hemin or Zn(II)PPIX binding, whereas the latter includes chemical shift changes for the P8 interaction. It is unclear whether these chemical shift perturbations are also observed when HusA is associating with hemin or Zn(II)PPIX. If there are chemical shift differences between hemin vs P8, it is necessary to specifically document these, as it may suggest conformational changes or additional porphyrin binding sites. Also, it is hard to be sure what is going on and if there are differences between even Zn-PPIX and hemin based on Supplementary Fig. 10.

Response: We thank the referee for highlighting a confusion regarding the effects of chemical exchange on the HSQC titration experiments with different porphyrins. To treat all porphyrin titration data consistently throughout the manuscript we have replaced old panels Fig. 2C and D, and replaced the old panel Fig. 4B with new panel Fig 5B using the same style of analysis for both – that is, including the changes in signal intensity as well as CSPs for each analogue. The raw HSQC data for the various titrations is now presented in Supplementary Figures S10, S11, S18 and S21. With this presentation it is clear that a similar subset of peaks is found in intermediate exchange (peaks that only disappear and do not return at molar excess of ligand), whereas other peaks are shifted from intermediate towards fast exchange when a different ligand is used. In other words, some peaks that appear firmly in an intermediate exchange regime with haemin and Zn-PPIX have shifted to a fast-intermediate regime with DPIX and P8, with the result that they are measurable further into the titration, and consequently larger CSPs can be observed.

We were interested to find signatures in the NMR data that could identify additional side group interactions, particularly with the larger P8 compound, however this was not evident. We suspect that bound ligands might adopt multiple poses within the same overall binding cleft and that these affects contribute to the observed chemical exchange and ‘smear’ out any specific signals from different porphyrin analogues. There are some qualitative differences between the different titration results, such as the more pronounced signal loss in the N- and C-termini when titrating with haemin (see Fig 2C). It is possible that these additional changes are due to paramagnetic iron effects, which must be present. We do not want to claim that the binding sites are identical, and the relevant sections of the text describe “similar set of residues” being effected in the titrations and that analogues bound to “similar sites”.

6) Binding stoichiometry: The authors have used a fixed [HusA] (1 μM) while varying hemin, DPIX, or P8 (Fig. 4C). If the curve fits were done correctly, they should not only provide K_d values, but also the ligand stoichiometry. To a first approximation, if one were to assume that DPIX interacts with a single site on HusA, then by extension it appears that there may be more sites for porphyrin binding when hemin or P8 are titrated with HusA. The manuscript is confusing in its use of the terms, such as "substoichiometric" (cf. p. 6, line 16). What exactly are the authors trying to achieve through the use of variable "molar equivalents" of porphyrin molecules?

Response: Substoichiometric amounts (amounts of ligand smaller than the number of available ligand binding sites) were useful in the NMR analysis to detect residues that were most affected by ligand interactions, and were naturally employed during standard NMR titration experiments. All of the NMR

titrations performed with different porphyrins showed a nucleus, or epicentre, of residues that were most effected at low concentrations of ligand and generally showed a graded response spreading out from this epicentre, consistent with a single predominant binding site, although we cannot rule out that other low-affinity, or 'non-specific' interaction sites exist for some or all porphyrins. Similarly, although the fluorescence titrations fit adequately to a 1:1 model we cannot rule out other more complex models.

Finally, concerns still remains with regard to hemin binding. The authors do numerous experiments including mutagenesis and comparisons to other proteins such IsdH but these are all indirect and fraught with difficulty by changing the heme environment. The authors still have not convincingly preformed any spectroscopy that could give a more detailed fingerprint such as resonance Raman or magnetic circular dichroism.

Response: To address this aspect of the manuscript we have sought expertise and Raman spectrometer access at Sydney Analytical Core Research Facility (directed by Prof Peter Lay). Unfortunately a recent laser upgrade of their custom-build J-Y Raman spectrometer resulted in numerous unforeseen problems with the mechanism to tune and guide the laser beam, which results in inconsistent data after numerous weeks attempting to acquire Soret-excited RR data. Nevertheless, using a different instrument (Renishaw InVia Raman microscope), we have recorded 488-nm-excited RR data for haemin and HusA and a Y164F mutant, and these results are presented in new Figure 3B. The three samples give highly similar spectra consistent with a 5c HS complex, and do not show features observed in some other Tyr coordinated haems. This supports our earlier assertion, based on ¹H NMR, UVvis spectra, acid-alkaline transition, and binding affinity, that there is no evidence to support Fe ligation in HusA.

Similarly, the binding of uroporphyrin and coproporphyrin does not show differences in spectrum. How was this shown to be specific?

Response: We agree with the referee and have changed the text here to read "HusA did not appear to bind the haem biosynthesis intermediates coproporphyrin III and uroporphyrin III" (Results pg 8, line 27). We have moved some speculative comments to the discussion "Although UVvis analysis did not indicate binding of coproporphyrin III, even a weak interaction could be physiologically relevant as coproporphyrin III (but not uroporphyrin III) can support *P. gingivalis* growth (Fig 1A and Supplementary Fig 3), consistent with a previous report that *P. gingivalis* lacks the enzyme to convert uroporphyrinogen III into coproporphyrinogen III but retains the enzymes that convert coproporphyrinogen III into PPIX and then to haem (Fig 1A)³⁰."

Furthermore, this work if all of the issues are addressed can stand on its own without resorting to discussions pertaining to the controversial "reverse ferrochelatase" activity, which I am unclear as to why the authors needed to include.

Response: Reference to the reverse ferrochelatase activity has been removed from Results. Some discussion of reverse ferrochelatase activity was requested by Reviewer 2, and so we have retained these comments in the discussion section.

Rigor, reproducibility, transparency, and presentation issues:

1) Why does the scaled fluorescence not approach zero in Fig. 4C post inner-filter effect corrections? One possibility is that the corrections have not been properly accounted for. This is evident in the green curve

shown in Fig. 3E. Have the authors accounted for Zn(II)PPIX having different fluorescence characteristics than the other porphyrins?

Response: There is no requirement for scaled fluorescence to approach zero in the scaling that we have chosen to perform. As stated in Methods, the scaling is done by setting the starting fluorescence to 1 and depending on the amount of quenching by the particular porphyrin and the concentration used. The final values may or may not approach zero. However, at a saturating concentration of porphyrin used, this should reach a plateau. For example, please see <https://academic.oup.com/nar/article/26/12/2955/2385888>
<https://www.ncbi.nlm.nih.gov/pmc/articles/PMC1796864/>
<https://academic.oup.com/nar/article/26/12/2955/2385888>

2) Re: Figs. 3E and 4C, it would be essential to provide not only the fits, but also a panel showing the goodness of fit, i.e. residual plot.

Response: These have been included in Fig. S14

3) SEM should include error bars for data in Fig 4C, which currently would be interpreted as being derived from a n=1 data set

Response: Please see response to Remaining Concerns 4) above.

4) Supplementary Fig. 3A and B appear to have been sliced and combined from different panels. If this is the case then the entire unmodified gel/blot should be shown in the Supplement for the sake of transparency.

Response: The original blot has been inserted in the new Supplementary Fig. 3A to replace the previous Supplementary Fig 3A and 3B.

5) What do the authors mean by ">= 2 standard deviations" in the context of Figs. 2D & 4B? Precisely, how were these calculated? It is ok to use the terms, such as "concave" and "convex", but it'd be important to point out to the reader that the former can be generated from the latter through a 180 degree rotation.

The effect of ligand binding on the peak height for each HSQC peak was calculated as $\log_e(\text{SN}_{\text{HusA_titration}}/\text{SN}_{\text{HusA_alone}})$, where $\text{SN}_{\text{HusA_titration}}$ and $\text{SN}_{\text{HusA_alone}}$ were the signal-to-noise in the presence/absence of the porphyrin. A significant change was arbitrarily set as a value of $\log_e(\text{SN}_{\text{HusA_titration}}/\text{SN}_{\text{HusA_alone}})$ more than 2 standard deviations lower than the average value of $\log_e(\text{SN}_{\text{HusA_titration}}/\text{SN}_{\text{HusA_alone}})$ across all the peaks in the spectrum. However, given that setting a threshold for 'substantial' signal changes is arbitrary, and the important point is to clearly identify the criteria and the threshold. The material and methods now read "The threshold for residues undergoing a 'large' change in peak height was set at approximately half the maximum peak height change in the spectrum, as shown by the black dashed lines in Figures 2C and Fig 4B"

The descriptions concave and convex refer to a shape with an 'inside' and 'outside' and hence they do not require an external frame of reference (i.e., they are unchanged by 180 degree rotation).

6) Title of Fig. 2 is misleading. It is not just the porphyrin binding site, but (as shown) hemin binding pocket as well.

Response: The title is now changed to “The structure of HusA and analysis of the porphyrin/haem binding site.”

7) What do the blue and red spheres represent in Fig. 2D?

These represented N and O atoms, originally to indicate the polar/non-polar nature of each side chain. We have removed this information from the new version of Figure 2.

8) Data should be included in Fig. 4F to illustrate the consequences of 1 uM P8, which W83 is susceptible to (p. 31, line 18). This data would provide the susceptibility index. Fig. 4F should be moved to the Supplementary Data.

Response: We thank the reviewer for this comment. A new Fig 4F including the data for susceptibility to 1 uM P8 is presented in the revised manuscript. As the reviewer recommended, this figure provides the susceptibility index for intracellular killing by P8. We consider it to be appropriate to present this information as a main text figure (Fig 4F) rather than as supplementary information.

9) What is the basis for the differences in the metronidazole conformations in the two conformers shown (Fig. 4B)? Is this a consequence of computational modeling or based on experimental data? This should be stated.

Response: As implemented by Autodock, the metronidazole compound has 19 rotatable bonds that sample different values during docking. Preparation of the ligands and docking procedure are described in detail in the section “molecular docking” in the supplementary methods. Figure Legend 5 now states “In silico docking of P8”

10) The authors should generate a similar figure as in 4B based on their chemical shift data, showing the interaction between P8 and HusA.

Response: The figure requested by the reviewer is now Figure 5B

11) What is the significance of Supplementary Fig. 7 where dynamics data is provided exclusively for the apo-protein. How do the parameters change upon P8 addition?

Response: The dynamics for the apo protein suggests that the apo structure is well ordered. These parameters could not be obtained for HusA in the presence of P8 or other porphyrins due to the effects of chemical exchange.

Mechanistic questions:

1) As administered, metronidazole is a prodrug and requires reductive activation for achieving its cytotoxic effects. Reductive inactivation of the nitro function results in the imidazole group cleavage. Alternatively, metronidazole is capable of interfering with chemiosmotic coupling and energy conservation. It'd be important for the authors to at least speculate on whether covalently fused metronidazole is capable of functioning like its free counterpart or if it is released? The experiments with

metronidazole plus PPIX or DPIX are not the same as with the P8 scaffold (including the Boc protecting group) plus metronidazole. As *P. gingivalis* is a heme auxotroph, would the authors expect P8 to impact other proteins?

Response: In related preliminary studies we obtained evidence that the conjugate remains intact but that the nitro group of the metronidazole moiety is reduced followed by cleavage of the imidazole ring. Accordingly, there is indication that the conjugated metronidazole appears to function similarly to free metronidazole. The enhanced potency and efficacy appears to be related mostly to uptake by the organism and particularly, to passage through the epithelial plasma membrane.

The other well studied haem scavenging protein of *P. gingivalis*, HmuY, coordinates iron by histidine and tyrosine. The iron ligand in HmuY plays a critical role to maintain the binding strength. However, we cannot exclude the possibility that P8 entry into *P. gingivalis* utilises the Hmu haem uptake pathway or other putative haem uptake systems. This is also demonstrated by the susceptibility of *P. gingivalis* to high concentrations of P8 in the absence of dipyrrolyl where HusA is barely detectable (Fig 4D).

As suggested by the Reviewer, we have edited the discussion section to include this information.

2) *P. gingivalis* is a nanoaerobe, do the authors expect P8 to block oxygen respiration? Are all the growth, viability, and potentiation assays performed under strict anaerobiosis? It appears to be the case, but the authors may wish to clarify whether or not O₂ contamination is likely during the course of the experiments. Notably, does *P. gingivalis* take up heme during anaerobic conditions or is it a consequence of triggering aerobic respiration.

Response: The growth and potentiation assays are prepared and conducted in the anaerobic chamber and leakproof screw cap test tubes. We have edited the methods section accordingly to clarify the experimental condition.

To our knowledge, *P. gingivalis* requires exogenous haem for growth due to the inability to synthesis the porphyrin tetrapyrrole *de novo* (Roper J et al, J Biol Chem. 2000, 22:275(51):40316-23; Smalley JW & Olczak T, Mol Oral Microbiol. 2017, 32(1):1-23). *P. gingivalis* acquires haem for iron and porphyrin is most likely taken up as an essential nutrient rather than as a consequence of aerobic respiration.

REVIEWERS' COMMENTS:

Reviewer #3 (Remarks to the Author):

The authors have taken a great deal of effort in improving and answering the concerns of the previous review, particularly with regard to the biochemical and spectroscopic concerns eg. resonance Raman in Fig 3 and the HSQC in Fig 4.

Similarly the inclusion of several additional controls in the growth assays tightens up the MIC data in Fig 1 and susceptibility data in Fig 4F

Overall the manuscript is greatly improved.